# Adaptive Dynamics of Settlement Models in the Urban Landscape of Termez (Uzbekistan) from c. 300 BCE to c. 1400 CE

**Enrique Ariño** [1,*] **, Paula Uribe** [2] **, Jorge Angás** [3] **, Raquel Piqué** [4] **, Rodrigo Portero** [1] **, Verónica Martínez-Ferreras** [5] **and Josep M. Gurt** [5]

1   Department of Prehistory, Ancient History and Archaeology, University of Salamanca, C/Cervantes s/n, 37002 Salamanca, Spain; rodrigoportero@usal.es
2   Department of Antiquity Sciences, University of Zaragoza, c. Corona de Aragón, 42 (Edificio Cervantes), 50009 Zaragoza, Spain; uribe@unizar.es
3   Department of Antiquity Sciences, ARAID—University of Zaragoza, c. Corona de Aragón, 42 (Edificio Cervantes), 50009 Zaragoza, Spain; j.angas@unizar.es
4   Department of Prehistory, Autonomous University of Barcelona, Edifici B Facultat de Filosofia i Lletres, 08193 Bellaterra, Spain; raquel.pique@uab.cat
5   ERAAUB, Department of Ancient History and Archaeology, University of Barcelona, Carrer de Montalegre, 6-8, 08001 Barcelona, Spain; vmartinez@ub.edu (V.M.-F.); jmgurt@ub.edu (J.M.G.)
*   Correspondence: argil@usal.es

**Abstract:** The archaeological site of Ancient Termez is located in southern Uzbekistan. Despite the arid environment, the city benefited from its strategic position near two rivers, the Amu Darya and the Surkhan Darya. Its significance was mainly related to the expansion of trade routes connecting Eurasia. The city comprises several enclosures that attest long-term human-environment interactions. In order to identify the adaptive dynamics of the settlement models during an extended chronology covering the period from c. 300 BCE to c. 1220 CE (Greco-Bactrian/Yuezhi, Kushan, Kushano-Sasanian, and Islamic periods), a multidisciplinary study has been carried out, which includes: (1) archaeological excavations in several areas of the urban complex; (2) pedestrian surveying inside some enclosures and in the urban periphery; (3) an aerial survey based on high-resolution satellite imagery; (4) AMS dating of charcoal and bone samples; (5) archaeobotanical investigation through anthracological analysis; (6) zooarchaeological studies. The results point to variations in the development of the inhabited spaces, in which abandonment and occupation took place. The zooarchaeological and archaeobotanical data demonstrate the exploitation of natural resources in different environments (i.e., arid areas and irrigated land) and a certain evolution during the period considered.

**Keywords:** archaeological survey; remote sensing; Anthracology; Zooarchaeology; ancient Termez; central Asia; ancient period; medieval period

## 1. Introduction

Evolution of settlement models in the city of Termez (Uzbekistan) between c. 300 BCE and c. 1220 CE is studied through data obtained with archaeological methodology. The information from successive excavation seasons at the site (from the 1930s to the present) has been complemented with data obtained in pedestrian surveying. The overall interpretation of the different enclosures in the city of Termez has combined the data in the archaeological record with information provided by CORONA and WorldView3 satellite images. The systematic analysis of the satellite data has identified superficial traces (visible in the micro-relief or differential growth of vegetation) visible in the urban layout of the antique and medieval city. Research on urban construction, destruction, and rebuilding processes in Termez has been completed with a study of the settlement's natural environment by examining the remains of plants (charcoal) and animals (bones) found in the excavations. The research carried out aims to incorporate the techniques of landscape archaeology in the study of the urban complex of Termez. Remote sensing and

surface surveys have been used to define the urban layout and the occupation chronologies of the different areas of the site. The results obtained with these techniques have been compared with the information provided by the archaeological excavations carried out at the site since 1936. A complete radiocarbon dating program has been used to obtain absolute dates in the use and reuse of the urban spaces. This has made it possible to address the dynamics of human occupation of the inhabited space between c. 300 BC and c. 1220 AD. The study also incorporates an analysis of the faunal remains and an important sample of charcoal recovered in the most recent excavation campaigns, in order to reconstruct the natural environment of Termez throughout the historical sequence. The final objective of the research is to determine how human activities and the natural environment were interrelated.

Termez was one of the most important urban centres of northern Bactria, a region of variable borders. It is generally accepted that during the Achaemenid Persian Empire (550–330 BC) its northern limit would have been marked by the course of the Amu Darya river, which would have delineated the border between the satrapies of Bactria and Sogdia. In the Seleucid period (323–238 BC), the border between the two satrapies would probably have shifted north of the Amu Darya, so that Bactria would have incorporated territories that, in the Achaemenid division, had been Sogdian. It is considered certain that at the time of the Greek kingdom (238–c. 140), Bactria would have incorporated into its dominions the region that today we conventionally call northern Bactria, which includes the lands north of the Amu Darya belonging to the republics of Uzbekistan, Tajikistan, and Turkmenistan. The northern boundary of the Greek kingdom would probably have been located in the gorge known as the Iron Gates, near Derbent (Uzbekistan). Little is known of the changes that could have been introduced by the nomadic Yuezhi Empire that ended the Greek kingdom of Bactria, but between the beginning of the 1st century AD and the middle of the 3rd century, the region was the central nucleus of the Kushan Empire, which far exceeded the limits of the ancient satrapy, especially to the east, as it included within its borders the city of Patna on the banks of the Ganges. The western part of the Kushan Empire—historical Bactria—was conquered by the Sassanian emperor Shapur I in military campaigns dated around 245–248. After the Islamic conquest of the region at the end of the 7th century, northern Bactria was included in the territory that Arab sources refer to as *Mawarannahr*, a term used to designate Transoxiana. Bactria is a geographical space with a dense urban occupation at least since the Achaemenid Empire. Classical sources refer to Bactria as the 'Land of a Thousand Cities' [1].

Old Termez was the predecessor of modern Termez. Both the old and new cities depend on the water in the Surkhan Darya, a tributary of the Amu Darya, which rises on the southern slopes of the Hisar mountain range (which reaches 4643 m in the summit of Khazret Sultan). In this way, irrigated crops can grow in an area characterised by its aridity. Old Termez was destroyed by Genghis Khan in 1220 [2] (p. 81) and [3]. The modern city is about ten kilometres to the south-east and was founded after the destruction of the old city. It was visited in the 1330s by Ibn Battuta, who described it as a prosperous town with numerous bazaars, irrigation canals, and gardens. Ruy González de Clavijo wrote about it in similar terms, after staying in the city in the autumn of 1404 [4] (pp. 8–11). Old Termez maintained a small population associated with craft workers until the 17th century [5] (pp. 438–439) and [6], but the place is now uninhabited.

The Termez archaeological site is at about 300 m above sea level in the Surkhan Darya Depression, which in turn is part of the Afghan-Tajik Depression (Figure 1). Some hills (Citadel, Kara Tepe, and Tchingiz Tepe) stand out on the plain with the ruins of the old city; these are the southernmost outcrops of the Cenozoic Baysuntau-Kugitangtau formation, a low mountain range (its highest peak is Khrebet Khaudat, 553 m) that delimits the Surkhan Darya basin on its western side [7] (pp. 92–93). The climate at Termez is characterised by a high evapotranspiration index (PETtho 1195.4) and marked aridity (Alu 0.12). The mean annual temperature is 16.5 °C. The warmest month is July (mean temperature 29.3 °C), and the coldest month is January (3 °C). The mean annual precipitation is

139 mm/y. Rainfall is distributed unequally, with most falling in spring, and varies greatly from year to year, from 78 mm/y to 223.4 mm/y. The lower valley of the Surkhan Darya is in the Irano-Turanian biogeographic region, with conditions suitable for low and not very dense xerophytic vegetation. The sudden temperature changes lead to freeze-thaw shattering of rocks and surface deposits. Winds are mainly from the south-east, with seasonal variations. South winds are most common in January, while east winds dominate in November [7]. The vegetation landscape of the region has been traditionally conditioned by human action. The lower Surkhan Darya valley is a traditional irrigated area, probably since prehistoric times. The Sherabad plain was not irrigated before the Soviet period. This area was put under irrigation in 1971 in order to develop large-scale cotton cultivation. The Sherabad plain is supplied with water by a large canal (the Sherabad canal) which takes water from the Surkhan Darya. Irrigation allows fruit tree crops, vineyards, and vegetables to thrive in the irrigated areas today, in addition to cotton, which still dominates large areas of cultivation [8,9]. The urban area of Termez is currently devoid of any tree cover, including the riparian vegetation that could thrive on the banks of the Amu Darya.

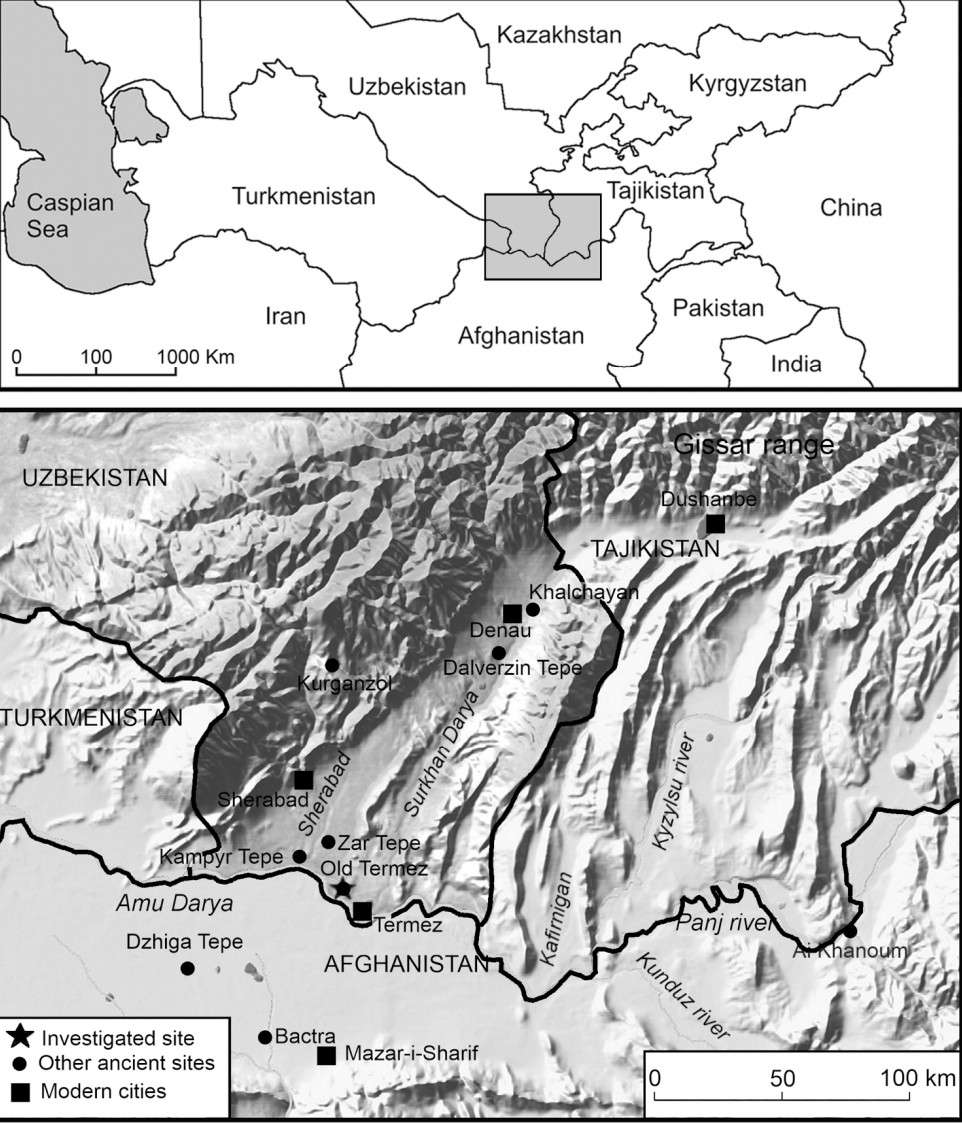

**Figure 1.** Map of Bactria with the location of Termez. Cartographic base 123RF (copyright: https://es.123rf.com/profile_mschmeling), accessed on 12 January 2022.

Wind and fluvial erosion are the two agents that have most affected the site's architectural structures and archaeological strata. Fluvial erosion has particularly damaged the Citadel and

Tchingiz Tepe enclosures in the westernmost part of the urban site. The first plans of the site in 1936 show that most of the southern part of the Citadel was missing at that time, although its rectangular shape was still recognisable [10] (pp. 126–130). Since then, a large part of the Citadel has disappeared because of erosion. The comparison of the CORONA satellite images (DZB00403800056H011001, 20 October 1964) and modern pictures on Google Earth is indicative of the impact of fluvial erosion in this part of the site in the last fifty years [11] (Figure 2). Wind and rain have also contributed to the loss of archaeological structures in ancient Termez. The impact of those agents can be observed by comparing the state of buildings and walls in images taken by Masson's archaeological expedition in 1936 [12] with their present appearance. Aeolian deposits are found in several areas of the site, especially in Tchingiz Tepe [13]. A geoarchaeological survey in Tchingiz Tepe showed that the surface deposits, which are over two metres thick in some places (2 to 2.8 m), have formed in the last 2000 years through the action of two natural agents, gravity and wind, and an anthropic process (construction, abandonment, and rebuilding of dwellings) [7].

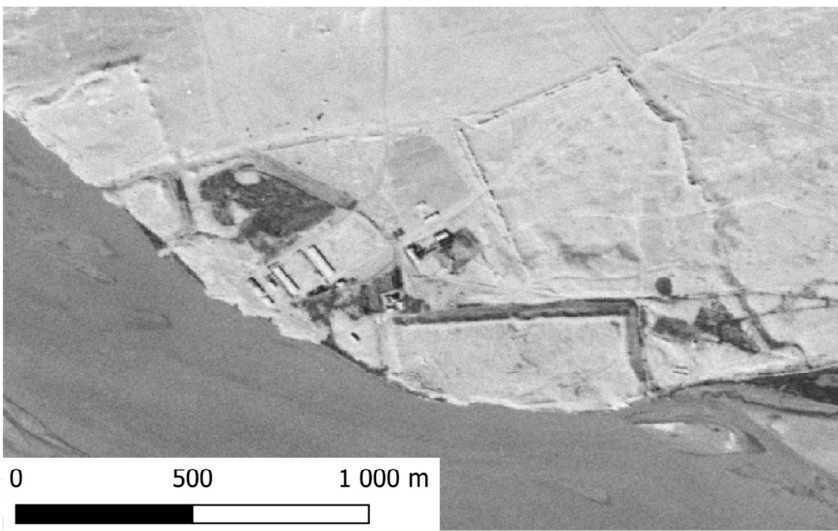

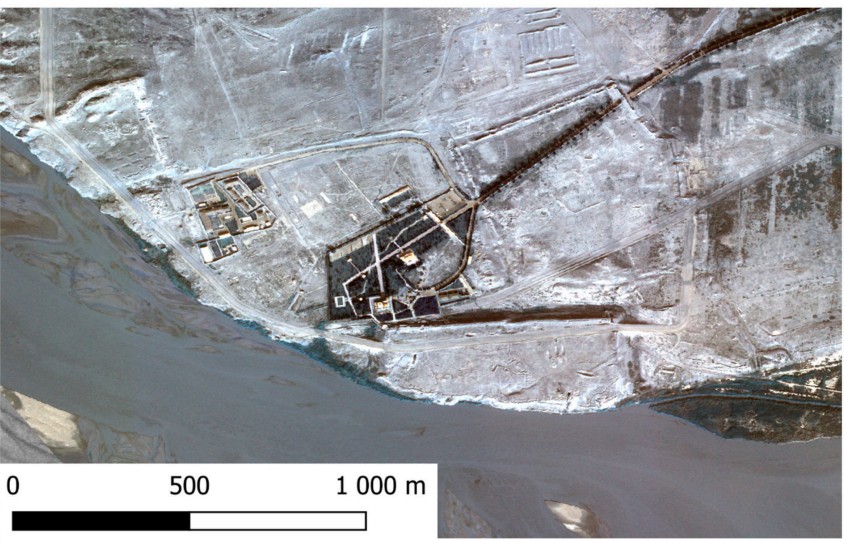

**Figure 2.** Comparison between satellite images of CORONA (DS1012-1039DA1963 20/10/1964) and WV3 (January 2017 012085673020).

The Termez archaeological complex occupies an area of about 550 ha. It comprises four fortified enclosures (Citadel, Tchingiz Tepe, Shahristan or Low Town, and Rabad). There was also human occupation in the suburbs, especially in the north (Potters' Quarter) and at

the site's eastern end, with a palace complex dated to about the 11th century, which has now disappeared (Figure 3). The oldest part of the settlement was in the Citadel (*Phrourion*), where archaeological levels are dated to the late 4th–early 3rd centuries BCE. These levels are associated with colonisation by Greek populations who were present in the region following the conquests of Alexander the Great [14,15]. Although the Citadel must have been founded as a fortified settlement, the original Greek wall has not been located. In contrast, the wall from the Kushan period, made with adobe, and the Islamic wall (of brick) are known; the latter was attached to the Kushan wall on its outer face [2] (pp. 83–97). Two large adobe buildings located 600 m north-west of the Citadel (Buildings A and B) have been interpreted as part of a cult complex. Excavations in this area have suggested an age in the Greco-Bactrian period, with some remodelling in the Kushan period [16] (pp. 190–200) and [17] (pp. 34–36).

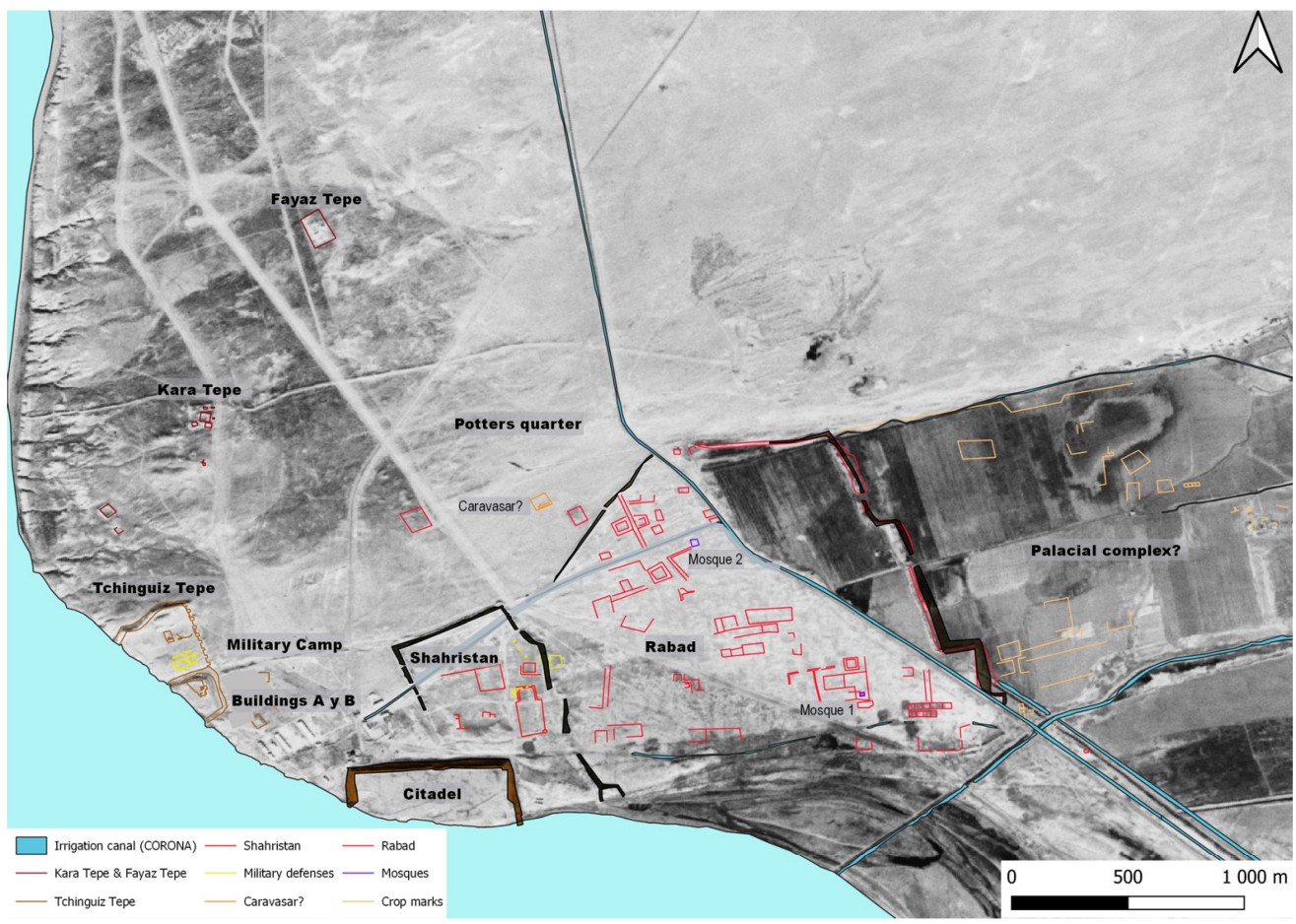

**Figure 3.** Image of Termez on the CORONA satellite (DS1029-2102DF114 09/02/1966) with the identification of the urban sectors mentioned in the text and the anomalies that may correspond to ancient urban structures.

Tchingiz Tepe's enclosure is located to the north of the cult complex. It has a trapezoidal shape, delimited by a wall of which only three sides have been preserved because of the action of the Amu Darya that has eroded the western wall. Tchingiz Tepe is the enclosure whose occupation sequence is best-known thanks to a full excavation programme accompanied by radiocarbon dating. The construction of the walls would have started in about the mid-second century BCE [13]. In the Kushan period, Tchingiz Tepe was the location of a Buddhist monastery [18]. The enclosure was no longer occupied after the 5th century, and it is the only fortified area in Termez with no evidence of medieval dwellings or activity.

The third walled enclosure at Termez is the Shahristan or Low Town. This district is located north of the Citadel and conserves the ruins of a thick wall of rammed earth (*pakhsa*). It is generally assumed that the wall was built in the Kushan period and rebuilt with the same technique in the Islamic period, although this hypothesis is based on field observation and the study of pottery in the rammed earth forming the wall [19,20]. Several parts of the Low Town have been excavated, but the results of this fieldwork have not been published completely. The only archaeological evidence supporting a Kushan presence in the Shahristan are some structures excavated in the bedrock, interpreted as Buddhist monastic cells, remains of a platform identified as part of a *stupa*, and a pillar capital decorated with acanthus leaves with an image of Buddha on each side [16] (pp. 189–190) [17] (pp. 50–55). The medieval Islamic sequence is much better documented and characterised due to the discovery of a group of houses made of stone and brick, many of them with blind wells used to drain wastewater. The foundation of these dwellings has been dated to about the 11th century, whereas their end is thought to be associated with the destruction of the city by Genghis Khan in 1220 [16] (188–189 and 210), Ref. [17] (pp. 44–45) and [21] (pp. 155–156). However, the Mongol conquest did not mark the definitive end of the occupation of this urban area, as several kilns used to fire pottery have been detected above the Islamic houses in the western sector of the Shahristan [22].

The Rabad, or Islamic City, is the largest of the walled enclosures at Termez, and it gives an idea of the vitality of the medieval city. The wall was built with *pakhsa* and was connected with the eastern part of the Shahristan wall. It enclosed a polygon with sides that contained defensive towers. Excavations have detected the ruins of at least two mosques. Mosque 1, in the south-east of Rabad, was founded in the 11th century and remodelled in the 12th–13th centuries. Mosque 2 was in the north of the Islamic city; its oldest phase is also dated to the 11th century, while it was remodelled on at least two occasions [23]. These remains documented by excavations in Islamic Termez are completed by a caravanserai, dated to the 12th century [24]. Outside the Rabad, at the site's eastern end, another walled enclosure held further important medieval remains. The TAKE expedition (Termez Archaeological Complex Expedition), directed by Masson in 1936–1937, found a palatial complex dated to the 11th century in this part of the city. No ruins of those buildings have been preserved because they were destroyed during the excavation of irrigation canals around Termez in the second half of the 20th century [4] (pp. 40–53).

Although the walled enclosures define the urban area, the city was also surrounded by its suburbs. On the plain between the enclosures of Tchingiz Tepe and Shahristan (Military Camp), our excavations have detected occupation levels whose oldest phase has been dated by radiocarbon to the Greek sequence (late 4th–early 3rd centuries BCE). The last phase in this sector is characterised by dwellings associated with the Islamic occupation. It does not display any evidence of destruction related to the conquest in 1220, although it should be borne in mind that only a small part of this area has been excavated [25,26] (p. 544).

To the north of the Shahristan and the Rabad, outside the walls, several potters' workshops have been documented in an inhabited area dated to the medieval period. For this reason, it is known as the Potters' Quarter [27]. Although this craft activity is undoubtedly well documented, the occupation sequence and urban structure in this suburb need to be defined more precisely. Finally, in addition to all these districts in the city, around Termez, the Buddhist monasteries of Kara Tepe [28,29], Fayaz Tepe [30], and Zurmala [31] (pp. 176–180), are all dated to the Kushan period.

In short, the historic layout of old Termez has been generally defined by archaeological fieldwork in several parts of the site, especially from detailed records of some of its sequences. Archaeological strata have been used to assign dates to the occupation of the walled enclosures, although the oldest phases in the different parts of the city are still largely undetermined. The functionality of the sites and the management of resources are equally pending study to a large extent.

## 2. Objectives, Materials and Methods

### 2.1. The Urban Dynamics in the Historical Sequence

One of the research objectives has been to analyse the dynamics of the construction, use, reuse, and abandonment of the different places in the city throughout its historical sequence. The study was designed to determine whether the urban expansion from the original centre of the Citadel really implied an increase in the inhabited area during the time in which the city was active (which would be indirect evidence of population growth) or if, on the contrary, the colonisation of new enclosures took place at the same time as the preceding inhabited areas were abandoned. The latter may have resulted from difficulties in maintaining the settlement areas due to the formation of natural aeolian deposits or anthropic causes (ruin and collapse of adobe architecture). Although archaeological field-work has been carried out in Termez since 1936, the publications are limited, and not all settlements can currently be identified in situ. Only some excavations in specific parts of the site have provided accurate stratigraphic records, while the date of the foundation of some enclosures still needs to be defined more precisely. Similarly, the date of the abandonment of the different parts of the urban site must also be better determined. Due to the large size of the inhabited area, intensive surveying and remote sensing were included in the study to combine the data provided by the archaeological excavations and surveys (surface finds of pottery) with the urban layout seen in satellite images.

#### 2.1.1. Intensive Surveying

Intensive surveying has been carried out in three of the least-known areas at the site. The Shahristan has been surveyed in two grids: Transect Sha1 (20 × 30 m) and Transect Sha2 (15 × 60 m). A single area of 15 by 45 m was surveyed in the Rabad (Rabad1). The transect KBTK1 in the Potters' Quarter covered an area of 30 × 60 m (Figure 4). This work was performed with surveyors a metre apart and, therefore, it can be considered that the whole surface of the transects was examined. In each area, all the potsherds visible on the surface were collected. These remains were georeferenced by sub-dividing each transect into squares 5 × 5 m in size. The visibility conditions were good in all the surveyed areas, with the squares in the Shahristan completely free of vegetation, whereas some areas in the Rabad were covered by light shrub vegetation that did not affect the apparent visibility. The wind acts on the sand, covering the upper part of the soil at the site, removing it, and building up deposits. However, the accumulations of sand are not deep enough to affect visibility. Many of the potsherds were affected by the erosional action of the wind, indicating that they were exposed on the surface for the time needed to generate those alterations.

The pottery assemblage gathered in the surface sampling was classified into classes or families depending on their formal characteristics. Greek pottery represents the oldest occupation in the sequence (c. 300–c. 140 BCE). It is defined by at least one of its surfaces being covered by a black slip. This is well documented in Termez [5,6,14,26] and at sites in the vicinity [32–37]. A new production, known as grey clay ware, appears in the ceramic record from the mid-second century BCE onwards. This pottery characterises the occupation associated with nomadic Yuezhi or Tocharian populations (also known as Early Kushan) as documented at Termez [5,6] and Dalverzin Tepe [35]. However, Mirzakultepe [38], in the immediate surroundings of Termez, is the site of reference for the characterisation and dating of this ware. The change of era and first decades of the 1st century CE (c. 50) are usually linked to the beginning of the ceramic production characteristic of the central Kushan period.

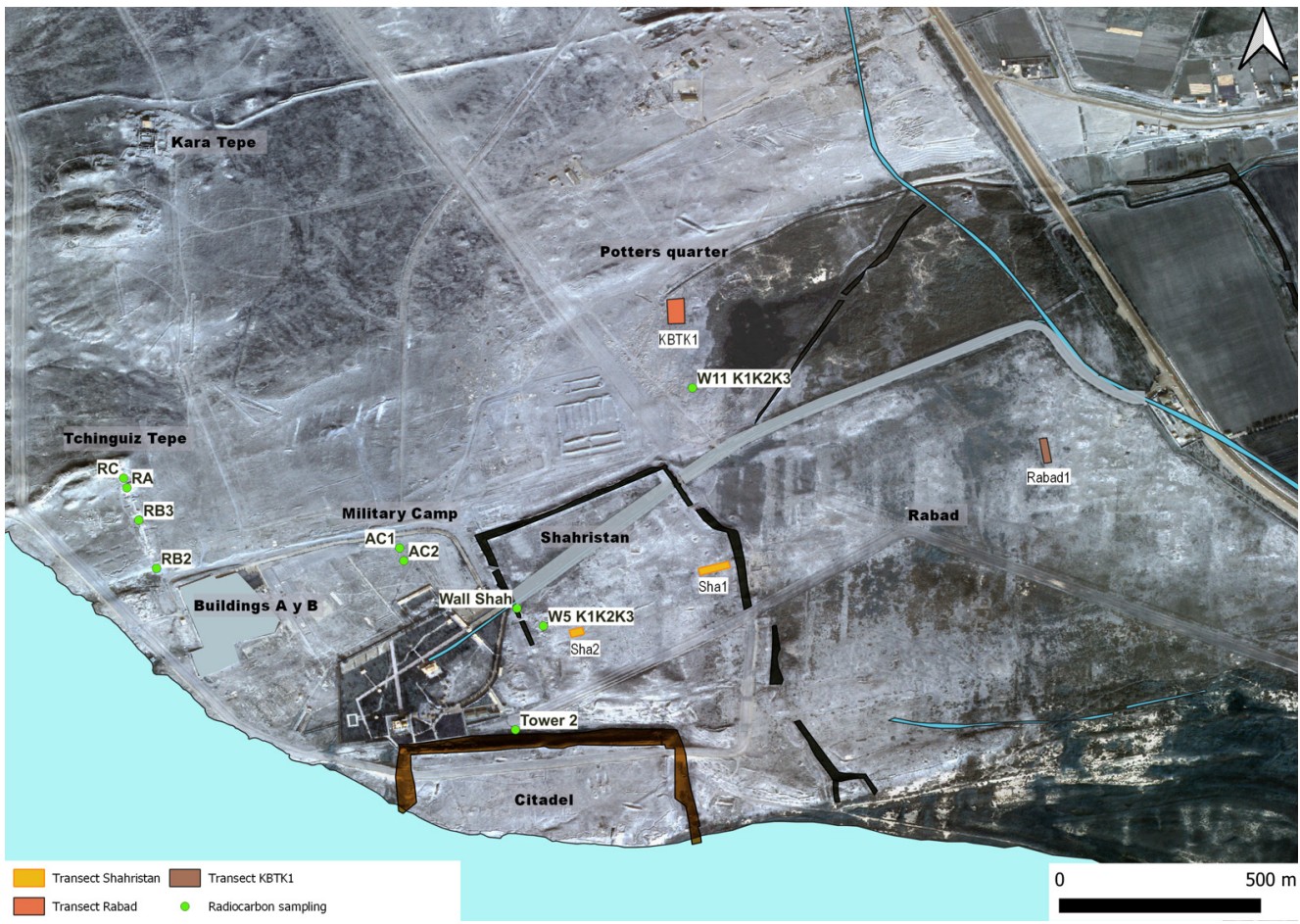

**Figure 4.** Image of Termez on the WV3 (January 2017 012085673020) satellite showing the location of the survey transects and radiocarbon sampling areas. 1: Transect Shahristan 1 (Sha1); 2: Transect Shahristan 2 (Sha2); 3: Transect Rabad (Rabad1); Transect Ceramics Quarter (KBTK1).

The most notable novelty is the use of red slips to cover one or both surfaces of the recipients. Although the deep red slip is the most typical cover of Kushan ware, other vessels were covered in orange, brown, or black (Figure 5). Kushan ware has been found at Termez [5,6,39] and places near the site, like Akkurgan [40], Dalverzin Tepe [35], Kampyr Tepe [41,42], Dzhiga Tepe [36], Aktepe, Shodmon Kala, and other settlements in the lower Kofarnihon valley [43,44]. In the mid-third century, the Kushan kingdom was subjugated by the Sassanian Empire. However, the pottery in the Sassanian period maintained the main features of the Kushan products, especially the red slips [45]; therefore, both sequences are grouped in the classification of sherds found in the surface surveying since stratigraphic data is not available to establish the sequence.

At the end of the 7th century, the region was conquered by the Arabs, resulting in a complete change in the formal and decorative repertoire of the pottery [5,6,22,27]. Glazes are frequent from the 9th and 10th centuries, especially in the case of tableware. They were often glazed with a single colour (white, green, yellow, or turquoise) but complex compositions with figurative or geometric motifs in different contrasting colours are quite common. The *sgrafiatto* technique consisted of scraping the surface with an awl before firing the vessel; it was combined with figurative or geometric decoration by applying layers of colour over the surface. Unglazed pottery is also common in Islamic pottery assemblage, usually decorated with various techniques, such as impression, incision, plastic applications, stamps, and moulds (Figure 6). Pots covered with a black slip or coating are also recognisable and exclusive to the Islamic period. This slip normally covers the interior

of the vessel and part of the outer surface in a pattern of drip lines from the rim to the base. Other characteristic productions of the Islamic period are spherical-conical vessels, often made in moulds and usually decorated with complex patterns (sometimes with Arabic letters). They have thick walls and a hole with a very small diameter which would have been used to control the pouring out of the product they contained. Despite being very common in the archaeological record of ancient Termez, their purpose is still unknown.

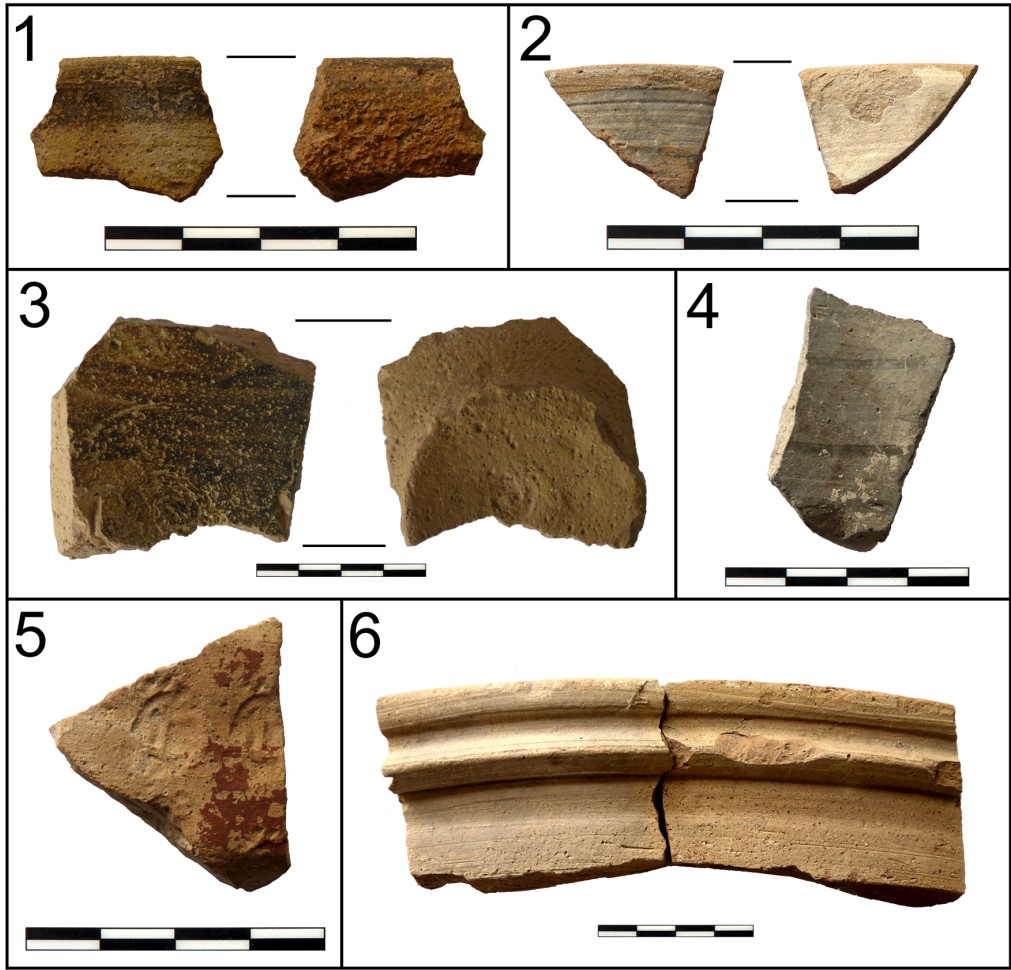

**Figure 5.** Ceramic productions recovered in surface surveying. Ancient period. 1: Red-figure ware (Greek) (Sha1); 2: White-slipped ware (Greek) (Sha1); 3: Black-slipped tableware (Greek) (KBTK1); 4: Grey pottery (Yue-zhi) (Sha1); 5: Slipped tableware (Kushan/Sassanian) (Sha1); 6: Common ware (Kushan/Sassanian) (Sha1).

Common ware and coarse cooking ware appear in all the historical sequences. These pottery types (the most abundant in the surface record) are generally regarded as non-specific and cannot be used to determine the occupation period in the surveyed area. Some of the sherds belonging to those products can be assigned to the Kushan and Kushano-Sasanian sequence if their degree of conservation is satisfactory enough to identify a characteristic shape, which is well-known thanks to typological classifications.

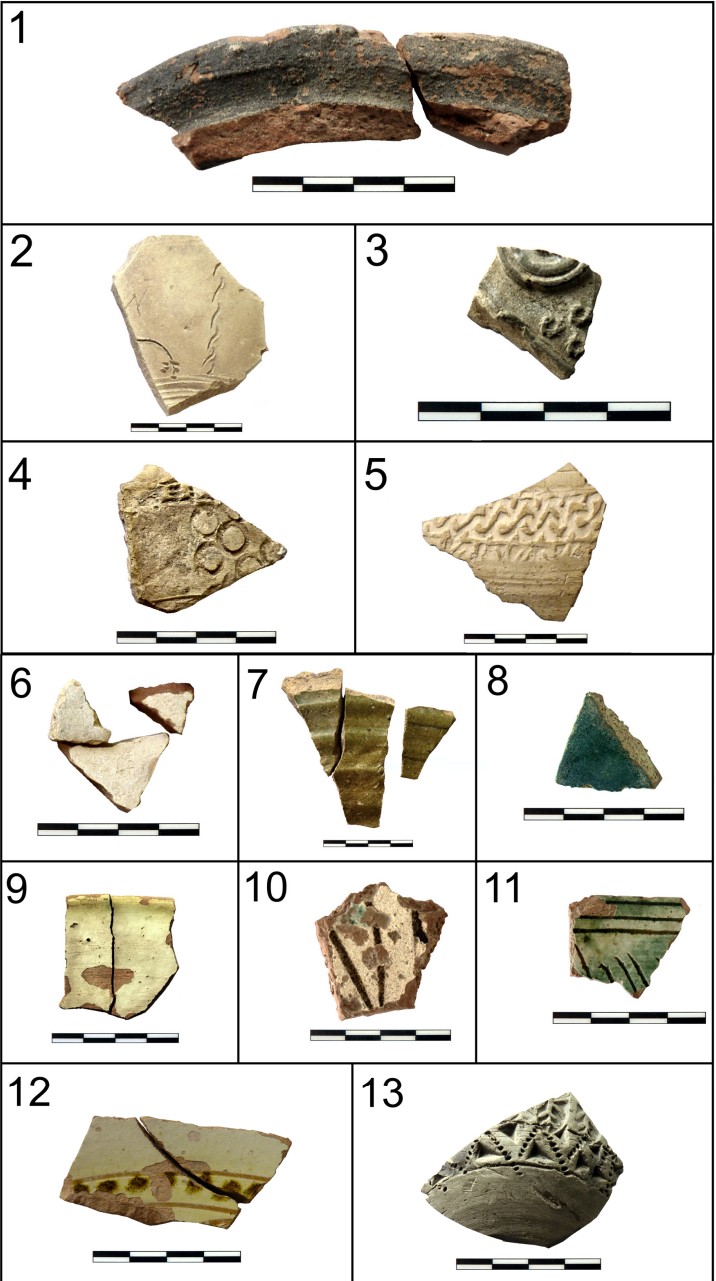

**Figure 6.** Ceramic productions recovered in surface surveying. Medieval period. 1: Common cooking ware, black slip partially covering the surface (Islamic) (Sha1); 2: Unglazed ware, comb-impressed/comb-incised decoration (Islamic) (Rabad1); 3: Unglazed ware, applied decoration (Islamic) (Sha1); 4: Unglazed ware, stamped relief decoration (Islamic) (Rabad1).; 5: Unglazed ware, moulded relief decoration (Islamic) (Sha2); 6: Glazed white ware (Islamic) (Rabad1); 7: Glazed green monochrome ware (Islamic) (Sha1); 8: Glazed turquoise monochrome ware (Islamic) (KBTK1); 9: Glazed yellow monochrome ware (Islamic) (Sha2); 10: Underglaze painted ware/slip painted (Islamic) (KBTK1); 11: Splashed *sgraffiato* ware (Islamic) (KBTK1); 12: Figured/geometric decoration glazed ware (Islamic) (Sha1); 13: Sphero-conical vessel (Islamic) (Sha1).

### 2.1.2. Radiocarbon Data

Samples of organic materials have been collected for radiocarbon dating in order to acquire new information about some of the less-known urban structures. A sample of bone was taken from the brick wall in the Citadel, and a second bone sample from the rammed earth wall in the Shahristan. Other samples were collected in the Islamic pottery kilns

in the western sector of the Shahristan. The radiocarbon samples were completed with others obtained in the systematic archaeological excavations in Tchingiz Tepe, the Military Camp and the Potters' Quarter. The radiocarbon results, which mark the dates of the initial occupation and the abandonment of each area in Termez, are given in Table A1. All the dates were calibrated with OxCal v. 4.4.4 [46] and the IntCal20 calibration curve [47].

### 2.1.3. Remote Sensing

Satellite programmes with a chronological difference of nearly 50 years (1967–2016), including sets of panchromatic and multispectral images, were used to study the landscape by remote sensing. The full analysis of all the data will require further post-processing and laboratory research in the future. However, the preliminary results are useful to identify the most appropriate sectors to carry out future fieldwork [11,48]. The comparative analysis of the data provides significant information about the area of study, which has suffered major changes in the last fifty years due to both natural and anthropic processes. The study was carried out with high-resolution space images in the CORONA programme (1967) and the commercial WorldView3 satellite images (from 2016–2017) by interpreting the images (seven of CORONA and two of WV3) with the best visualisation of the remains. We have mainly worked with the images DZB00403800056H011001 in the CORONA programme and two in the WV3 (16AUG29063625-M2AS R1C2-012085673010 01 P001 from 2016 and 17JAN1305145-M2AS R1C2-012085673020 from 2017). Other CORONA images were examined to improve the visualisation of specific areas. The old city of Termez is in a natural and political border zone which is still of strategic interest, and therefore systems that might provide better resolution (such as drones) cannot be used. There is no automatic method for the pre-treatment and post-processing of the images and, therefore, the different processes used with the two satellite image programmes are described.

The first step is to determine both programme's world covers to purchase the images. Among the CORONA programmes, we selected images dating from 1960, known as DISP (Declassified Intelligence Satellite Photographs) [49], representing one of the oldest digitalised periods with a satellite sensor. In recent years, CORONA has been used by archaeologists in the Near East [50–52] because the various declassifications (1996, 2002, and 2013) offer a practically global cover with high-quality images and stereoscopy in some missions. In many cases, the images show the situation before the urban and agricultural changes and construction of infrastructures that took place from the late 1970s; therefore, they provide information of exceptional quality. In Termez, the images show the state of the landscape before the Afghan–Russian war and the anthropic changes that occurred in more recent times. After their purchase, the CORONA images were subject to a single treatment: georeferencing with the UTM coordinate system, Zone 42 (EPSG: 32642) and the WGS 84 datum, by referring to points in common with the WV3 images. The spectral resolution of the CORONA images varies, with different filters for the reduction of contrast [53], but it is always within the visible range for their panchromatic condition. As a result, it was not possible to generate neo-channels, which was possible with the WV3 images.

By complementing this CORONA programme with WV3 images, it was possible to extend the temporal framework to 2016–2017, increase the spatial resolution, and obtain multi-spectral information, which is not possible with the CORONA programme. The WV3 images offer greater spatial and spectral resolution, with panchromatic images and eight multi-spectral bands, with a pixel size of $0.31 \times 1.24$ m. The pre-treatment of these images is more complicated than in the case of the CORONA series. Once acquired, it is necessary to conduct an atmospheric correction, georeference them (UTM WGS 84, Zone 42, EPSG: 32642), and obtain a mosaic with a final pan-sharpening.

The atmospheric correction was achieved by rectification of the reflectivity values through a comparative statistical analysis of the results of the terrestrial cover with values of a Landsat-8 OLI image of a similar date. Homogeneous areas, such as bare land, pastures, shrubland, crops, water, and asphalt, were selected. The extraction of the mean value and the standard deviation of the WV3 pixels (the latter to verify the disparity in the values in

a Landsat pixel) were performed with QGIS software. This process generated a database with information on 30 control points in the different land cover in the images.

The reflectivity values nearest to the true-earth values were identified with: (1) the Shapiro–Wilk test to determine the normality in the distribution of the variables; (2) ANOVA analysis; (3) the R2 determination coefficient; and (4) the root square mean error (RSME). The spectral signatures of each point were also analysed visually.

After obtaining the reflectivity image closest to reality, a pan-sharpening process was performed by combining the multi-spectral image with the panchromatic image obtained simultaneously to improve the visual interpretation of the photo-interpreted elements. The methods used were: Brovey Transform, Subtractive Resolution Merge (SRM), High Pass Filter Transformation (HPF), substitution based on the use of the ACP transformation, and substitution based on the HIS transformation [54]. These were all carried out with the Erdas Imagine 2015 software. Finally, the HPF method was selected to carry out the secondary magnitudes. Despite not being the image with the best colour, it achieves a very good spectral and spatial resolution and is a method with well-balanced results.

With the high-resolution spatial images from January and August 2016–2017, the pan-sharpening process was used to create secondary magnitudes. Neo-channels were created with the Erdas Imagine 2015 software for the most common indices, such as NDVI, SAVI and SR, and the ACP and edge detection techniques. The other indices (GRVI, DVI, RDVI, GDVI, GNDVI, IRG, MTVI, RDRER, GARI, OSAVI, GEMI, MSR, and RGRI) were obtained with the ArcMap 10.5 raster calculator to combine the bands resulting from the pan-sharpening process. The treatment of these indices has complemented the data identified with the CORONA programme. Finally, the structures visible in each index were drawn following the indicators of soil and crop marks.

### 2.2. Vegetal Landscape

Our excavations in Tchingiz Tepe and the Potters' Quarter have retrieved a significant number of charcoal fragments, which were examined to study the vegetal landscape in Termez and its surroundings at particular times in its historical sequence. A total of 220 fragments have been analysed from Tchingiz Tepe. These were selected by hand during the excavation of several structures in the sectors RB2 (SU 6, collapse level), RA (SUs 23 and 36, collapse levels), and RF (SU 26, the fuel used in a pottery kiln). The 57 samples from the Potters' Quarter were recovered during the excavation of Workshop 11 (the foundation level of the workshop on the bedrock, Kiln 1, and the waste dump that filled a *tannur* in one of the dwelling areas). The taxonomic classification of the archaeological charcoal was achieved by studying its microanatomy and comparing it with a reference collection. The anatomical analysis was performed with a reflected light microscope equipped with several lenses allowing $40\times$ to $400\times$ magnification. Identification of the taxon was based on observations of three sections of the wood (transversal, radial longitudinal, and tangential longitudinal) obtained by manually splitting the charcoal fragment. The keys used in the identification were published by Schweingruber [55], while the comparisons were made with the charred wood reference collection from the Archaeological Analysis Service at the Autonomous University of Barcelona. The charcoal corresponds to two sequences. The remains obtained in Tchingiz Tepe belong to Antiquity (2nd–5th centuries CE) and those from the Potters' Quarter to the Middle Ages (8th–10th centuries). These ages are based on the study of the stratigraphic record in those areas and supported by radiocarbon dates.

### 2.3. Fauna

Information about the fauna in old Termez has been obtained by studying skeletal remains found in the excavations conducted in six areas of the site: Tchingiz Tepe (sectors RA, RB1, RB2, RB3, and RC; Military Camp (sectors AC1 and AC2); Shahristan (Workshop 5); pottery kiln in Kara Tepe (Workshop 6); and Potters' Quarter (Workshop 11). The archaeozoological study has been carried out by grouping the finds in cultural sequences. The first is the Greek and Yuezhi periods (c. 300 BCE to 50 CE); the second covers nearly

the whole Kushan and Kushano-Sasanian sequence (c. 150–600 CE), although data from the earliest phase (c. 50–150 CE) is missing. Practically all the information for the medieval period corresponds to the time after the Islamic conquest of the region (c. 800–1400).

The skeletal remains were studied with standard archaeozoology and taphonomy methodologies [56,57]. The anatomical and taxonomic identification of the remains was based on different osteological atlases for domestic and wild animals [58–61] and the reference collections of the Aranzadi Science Society in San Sebastián, Spain, and the Archaeoscience Laboratory in Lisbon. Several guidebooks on the fauna of Central Asia were also used to identify the species [62–65]. Specific studies were consulted to differentiate between species in a family or group of taxa [66–75]. Similarly, osteometric criteria were considered to discriminate between the domestic species represented [70,76–79]. To quantify the representation of skeletal elements, the Number of Remains (NR), Number of Identified Specimens (NISP), and Minimum Number of Individuals (MNI) were calculated by taking into account the most numerous anatomical part, the laterality of the bone, and the age and sex of the animal [56,57,80,81]. When the remains could not be identified to species or family level, they were grouped into three size categories, large (>200 kg), medium (100–200 kg), and small (<100 kg). The full data obtained in the faunal study at Termez can be consulted in Portero et al. [82,83].

## 3. Results

### 3.1. Changes in the Use of Inhabited Areas

3.1.1. Chronology and Occupation of Urban Spaces

The 11,899 potsherds found by surface surveying correspond to the following samples: Shahristan 1 (Sha1, $n$ = 4674); Shahristan 2 (Sha2, $n$ = 4014); Rabad (Rabad1, $n$ = 1246); and Potters' Quarter (KBTK1, $n$ = 1684). Table A2 quantifies the sherds in each of the transects of the survey and also their cultural ascription in order to allow a comparative analysis between the sequences. Common ware is the most frequent type in all the samples but cannot be used to identify the occupation as, in most cases, it is impossible to determine its cultural ascription.

Greek pottery appears in small quantities, only in Sha1 and KBTK1, and it can be considered off-site material. Human presence in the Seleucid and Greco-Bactrian periods is documented in the Citadel [14] and Tchingiz Tepe [13]. Therefore, the presence of Greek pottery in the Shahristan and Potters' Quarter may be attributed to activity in places close to the areas that were really inhabited. The small sample of Greek pottery includes a fragment with red figures and one with a white slip in its interior. The occupation sequence in the Tchingiz Tepe enclosure is well documented by a complete excavation programme and $^{14}$C dates. At least three dates support human activity in the enclosure in the 2nd and 1st centuries BCE (Table A1: Tchingiz Tepe, Sector RB1, SU 5; Tchingiz Tepe, Sector RB2, SU10; Tchingiz Tepe, Sector RB2, SU11). However, the time of construction of the wall is still uncertain since the calibrated dates cannot discriminate between the time of the Greek kingdom of Bactria, or the Yuezhi period.

The surface survey results in the Shahristan indicate that the human occupation there can be dated to the Kushan period. Pottery associated with the Kushan and Kushano-Sasanian sequence is abundant in both sectors of the survey, especially in Sha2 ($n$ = 382, 9.52% of the total), where it even outnumbers Islamic pottery ($n$ = 346, 8.62%). In Sha1, Islamic pottery is more abundant ($n$ = 655, 14.01%), but the number of sherds supporting the older period is still significant ($n$ = 167, 3.57%). Excavations in the Shahristan have found remains consistent with a Kushan occupation [16] (pp. 189–190) and [17] (pp. 50–55). A radiocarbon date on a bone taken from the western part of the Shahristan wall corresponds to the time between the change of era and the early 2nd century CE (Table A1: Shahristan, Wall), which is further evidence for human occupation in that enclosure starting in the Kushan period. Moreover, data in other surveyed areas suggest that human activity expanded from the original centre in the Citadel over the whole urban area. Both the Rabad and the Potters' Quarter have provided a large number of potsherds dated to the Kushan

and Kushano-Sasanian periods (*n* = 67, 4.39% in the Rabad; *n* = 39, 2.32% in the Potters' Quarter). In fact, the percentages of this type in the Rabad survey are higher than those in Sha1, so a Kushan population in that part of Termez cannot be ruled out even if no archaeological structure of this period has been detected to date through excavations [23,24]. The suburban area known as the Military Camp, located between Tchingiz Tepe and the Shahristan, was also inhabited in the Kushan period. Archaeological excavations have recovered pottery consistent with a stable occupation and determined radiocarbon dates that frame the oldest occupation phase in the 1st century CE, although an earlier occupation cannot be completely ruled out (Table A1: Military Camp, Sector AC1, SU12; Military Camp, Sector AC2, SU20; Military Camp, Sector AC2, SU33 [25] and [26] (p. 544).

The surface survey shows that the Islamic occupation was intense in all sectors, which is consistent with the excavation data. Two mosques and a caravanserai have been documented in the Rabad [23,24]. Similarly, the Potters' Quarter was used intensely in the Islamic period, with both potters' workshops and dwellings (Table A1: Potters' Quarter, Workshop 2, Kiln 2; Potters' Quarter, Workshop 11, *Tannur*; Potters' Quarter, Workshop 11, Kiln 1, SU1; Potters' Quarter, Workshop 11, Kiln 1, SU1; Potters' Quarter, Workshop 11, Kiln 3, SU21; Potters' Quarter, Workshop 11, SU32) [22,27,84]. The Shahristan was used for dwellings in the Islamic period [22,27]. In the Military Camp, the absolute dates and the stratigraphic record support an Islamic occupation (Table A1: Military Camp, Sector AC2, SU12; Military Camp, Sector AC2, SU18). Only Tchingiz Tepe seems to have been uninhabited in that period. Radiocarbon dates for Tchingiz Tepe indicate continuous occupation from its origin in the Greek or Yuezhi period (Table A1: Tchingiz Tepe, Sector RB3, SU2; Tchingiz Tepe, Sector RB3, SU6; Tchingiz Tepe, Sector RC, SU10; Tchingiz Tepe, Sector RF, SU20) but the most recent dates suggest that human settlement in the area did not continue after the mid-fifth century (Table A1: Tchingiz Tepe, Sector RA, SU17; Tchingiz Tepe, Sector RC, SU2; Tchingiz Tepe, Sector RC, SU3; Tchingiz Tepe, Sector RC, SU4; Tchingiz Tepe, Sector RC, SU5). The stratigraphic record in Tchingiz Tepe does not include a single level corresponding to the Islamic period [13]. The destruction of the city by the Mongol army of Genghis Khan affects the whole urban space. A $^{14}$C date for the Citadel wall seems to be associated with building work or at least a repair in the late 11th or early 12th centuries (Table A1: Citadel/Tower 2). In the Shahristan, the pottery kilns have been dated to the late 14th or early 15th century (Table A1: Shahristan, Workshop 5, Kiln 3); however, this sequence seems to be residual as the potters' workshops are located in what had been a dwelling, which can be interpreted as evidence of the loss of the residential use of that part of the city.

### 3.1.2. Town Planning

Archaeological levels corresponding to the oldest phases in the Citadel are covered by thick medieval deposits, consisting of several metres in some areas. In this situation, the detection of evidence of structures belonging to the Greek, Kushan, and Sassanian periods through the analysis of satellite images is unlikely; however, it was able to provide important information about the Islamic occupation phase. An exception is the Citadel, where no type of urban structure appears, not even associated with its final occupation in the early 13th century. The 2017 image only reveals traces related to the use of the area for military training (Figure 3).

The information obtained by excavations, intensive surveying, and radiocarbon dating demonstrates that the Shahristan was occupied from the Kushan period to at least the 14th–15th centuries; therefore, the structures detected by remote sensing may belong to any of these periods. The study has detected traces that can be associated with the ruins of a structure of unusual size and characteristics (Figure 3). It is large and rectangular (206 × 111 m) and includes a rectangular head or annexe on its northern side (77 × 62 m). It is perfectly visible in the CORONA images 114_d/1966; 114 c/1966; 004c/1965; and 001c/1975. However, in the modern WV3 images (2016–2017), only the southern side of the structure can be observed in the January GARI vegetation index. The differences

between the two satellite images show the importance of using multi-temporal data in remote sensing analysis.

It is hard to imagine the function and chronology of the structure visible in the CORONA image, but its shape, size, and apparent lack of internal divisions suggest that it might be interpreted as a religious or palatial space, perhaps associated with the Kushan period, a time when large monumental constructions with regular floor plans were built in Bactria. Some possible parallels are the Temples B and C at Surkh Kotal [85], the so-called temple of the Dioscuri in Dilberdzhin [86], and the palatial enclosure at Khalchayan [87]; the degree to which the structure in the Shahristan resembles the latter is much larger. An attribution to the Kushan period is also supported by the fact that the building does not seem to correspond to the typologies of the madrasahs or mosques detected in other parts of Termez associated with the Islamic occupation.

Many anomalies, visible in the old CORONA photographs (especially in the February 1966 image) and the more recent WV3 photos, are observed inside the Rabad enclosure. One that appears in the northern part stands out; in this area, there is now a small lake that formed barely fifty years ago, and which hinders archaeological exploration. The CORONA images provide information about how this northern part of the Rabad may have been organised. The 1966 image shows a large street in a north–south alignment. Two symmetrical spaces 50 × 55 m in size are associated with the middle part of the street. These two rectangular areas are identified as a topographic anomaly formed by the collapse of the perimeter walls. The relief that marks their position in the terrain is clearly visible in the 1964, 1966, and, above all, in the July 1975 photographs. The interpretation of these structures is difficult, although they would not correspond to propylaea, like those at Ai Khanoum, as the Termez structures are much larger (the structures at Ai Khanoum are 7 × 9 m in size) [88]. What is clear is that this street and the rectangular shapes annexed form a monumental complex around which the whole urban layout of the north sector of the Rabad is organised.

Judging by the traces visible in the satellite images, the layout of the Rabad was organised on a regular premeditated plan, although with variations in the directions of the streets. In the south, the alignment of the streets and buildings differs from those around the large street that articulates the north. The study of anomalies allows identifying the internal differences that might correspond to functional divisions. For example, in the north-east of the Rabad, some structures might be identified as large buildings with complex floor plans, probably residential structures. In contrast, the ruins in the south-west of the urban space are simpler and smaller, which might be indicative of an artisans' quarter in that sector. It is likely that the regular town planning discernible in satellite images corresponds to the Islamic occupations. Previous excavations in the Rabad identified two mosques, one of them built next to a dwelling [23], and they are visible in the satellite photographs. The southern mosque is perfectly visible in the WV3 panchromatic image from 2017 but not in the CORONA. It appears oriented in the same direction as the other traces in that part of the enclosure; in contrast, the mosque in the north is inserted in the regular layout around the monumental street in that area.

A caravanserai dated to the 12th century was documented in the Rabad [24] and excavated in 1986. This construction is no longer visible on the surface and does not appear in CORONA images. In contrast, the WV3 image exhibits some traces in the ground surface that may correspond to the ruins of this building. They are located north of the Rabad, outside the enclosure, near the north access. The palace to the east of the Rabad, excavated by the TAKE expedition in 1936 and 1937 [12], does not appear in the satellite images, although the area has been significantly altered since that time by irrigation channels for farming.

The TAKE expedition, directed by Masson in 1936–1937, excavated the palatial complex outside the Rabad, on the eastern side of the site, and dated it to the 11th century. This is now an area of crops, where the vegetation indices obtained with the WV3 images ought

to detect traces of the ruins. However, the structures cannot be identified, perhaps because of the natural or anthropic sediment deposition [12].

*3.2. Vegetal Landscape*

The anthracological sample corresponding to the Kushan and Kushano-Sasanian sequence (2nd–5th centuries CE) comprises 220 pieces of charcoal. Their study reveals a predominance of taxa that thrive in riparian woodland and wetlands (Table 1, Figure 7). Trees and shrubs associated with riverbank vegetation are represented by willow and/or poplar (Salicaceae, *n* = 69) and tamarisk (*Tamarix* sp., *n* = 13). These species can be found in alluvial plains and in river valleys in Central Asia known as *tugai* [89,90]. The oleaster (*Elaeagnus* sp., *n* = 26) is native to the central Asian steppes and also flourishes in riparian sites, although it also tolerates moderately dry ground. Oleaster fruit can be consumed, and the tree might be encouraged or manged by traditional human societies. Monocotyledoneae (*n* = 32) represents many herbaceous species including steppe grasses and species that thrive in more humid conditions. Some charcoal was identified as juniper (*Juniperus* sp., *n* = 4), a genus with several slow-growing species that currently thrive in the region in dry and montane environments. Others, such as elm (*Ulmus* sp., *n* = 3) also grow in mountain areas in the region, which suggests that wood was acquired in more distant places. The plane tree (*Platanus orientalis*, *n* = 55) and nettle tree (*Celtis australis*, *n* = 1) are species that are thought to have been introduced into the region and therefore their presence in the site is indicative of a very early introduction and they may have been planted intentionally. Similarly, the common grapevine (*Vitis vinifera*, *n* = 5), the *Prunus* genus (*n* = 1), which may correspond to such fruit trees as cherry, plum, peach, apricot, and almond, and other remains identified as the Maloideae sub-family (Rosaceae/Maloideae, *n* = 1) suggest the possibility of cultivated species. Although it is not possible to discriminate between cultivated and wild varieties from the wood anatomy, other archaeobotanic studies in Central Asia, along the Silk Road, have documented the growth of grapevines, almonds, and peaches [91–93]. They might indicate the use of water resources to grow domestic species that would not flourish in the arid environment of Termez without a supply of water. They are all species grown today in the orchards of Termez thanks to irrigation with water from the Surkhan Darya.

Most of the pieces of charcoal associated with the Kushan and Kushano-Sasanian sequence (2nd–5th centuries CE) correspond to branches of a small calibre (up to 20 mm or a little larger), which seems to indicate selective gathering of wood, probably to be used as fuel, although those types of branches might be used for buildings, especially as roofing material. Pruning would have allowed the use of trees and shrubs for either of those two purposes. In the case of plane trees (*Platanus orientalis*), evidence has been found that the branches were cut in spring. The only taxon of which trunks have been identified is Salicaceae (willow or poplar), which suggests that the wood of this family was used as building material or sometimes to make furniture and tools.

The medieval sample (8th–10th centuries CE) was smaller, with 57 pieces. Nine riparian species were documented: willow or poplar (Salicaceae, *n* = 12), tamarisk (*Tamarix* sp., *n* = 11), and ash (*Fraxinus* sp., *n* = 7). Oleaster (*Elaeagnus* sp., *n* = 3) and *Prunus* sp. (*n* = 2) also appear. Nine pieces were identified as the Leguminosae family, which includes three sub-families: Caesalpinoioideae, Mimosoideae, and Papilionaceae. Their anatomic characteristics allow them to be attributed to the Papilionaceae sub-family, but the anatomical similarities hinder their identification at the taxonomical level. One piece of charcoal may be classified in the Chenopodiaceae family, which includes over 1500 herbaceous plants. Finally, as in the old phase, Monocotyledoneae has been documented (*n* = 3); it is a class that encompasses all herbaceous species. The lesser taxonomic diversity in this period might result from a smaller sample. However, continuity is seen in the use of riparian vegetation, especially Salicaceae and tamarisk, which are still predominant, together with ash. Again, most of the charcoal remains come from branches with a small calibre, which indicates a clear selection of that type of wood, possibly in connection with the function of the source of the sample, which is linked to a *tannur*.

Few anthracological studies have been carried out at archaeological sites in Central Asia to put the results of the present study in context. The use of Salicaceae as fuel was documented in Mesolithic and Bronze Age levels at Aigyrzhal-2 and the Naryn valley in Kyrgyzstan [94], and it has also been documented at 6th–4th century BCE sites in Kyzyltepa in Uzbekistan [95]. Despite the chronological and spatial distances, the continuous use of riparian woodland for firewood is worth noting.

**Table 1.** Frequencies of the different vegetal taxa documented in Termez. Indeterminate means that, although the anatomical features are well observed, we have not been able to identify it due to lack of reference materials. Not determinable means that the anatomical features have not been well preserved and do not provide enough information.

| | Ancient Period (2nd–4th AD) | | | | |
|---|---|---|---|---|---|
| **Taxa** | **Tchingiz Tepe/Sector RB2 SU 6** | **Tchingiz Tepe/Sector RA SU 23** | **Tchingiz Tepe/Sector RA SU 36** | **Tchingiz Tepe/Sector RF SU26** | **Total** |
| *Celtis australis* | 1 | - | - | - | 1 |
| *Elaeagnus* sp. | 16 | - | - | - | 26 |
| *Juniperus* sp. | - | 1 | 3 | - | 4 |
| Monocotyledoneae | - | 11 | 1 | 20 | 32 |
| *Platanus orientalis* | 31 | 2 | 22 | - | 55 |
| *Prunus* sp. | - | 1 | - | - | 1 |
| Rosaceae/Maloideae | - | - | 1 | - | 1 |
| Salicaceae | - | 37 | 32 | - | 69 |
| *Tamarix* sp. | - | 5 | 8 | - | 13 |
| *Ulmus* sp. | 2 | - | 1 | - | 3 |
| *Vitis vinifera* | - | 5 | - | - | 5 |
| Indeterminate | - | 1 | 2 | - | 3 |
| Not determinable | - | 4 | 3 | - | 7 |
| Total | 50 | 75 | 75 | 20 | 220 |
| | Medieval period (8th–10th AD) | | | | |
| Taxa | Potters quarter/*tannur* | Potters quarter/Kiln 1 SU 1 | Potters quarter/foundational level on bedrock | | Total |
| Chenopodiaceae | - | - | 1 | | 1 |
| *Elaeagnus* sp. | - | 2 | 1 | | 3 |
| *Fraxinus* sp. | 2 | 5 | - | | 7 |
| Leguminosae | 7 | 1 | 1 | | 9 |
| Monocotyledoneae | 3 | - | - | | 3 |
| *Prunus* sp. | - | 1 | 1 | | 2 |
| Salicaceae | 12 | - | - | | 12 |
| *Tamarix* sp. | 11 | - | - | | 11 |
| Indeterminate | 9 | - | - | | 9 |
| Total | 44 | 9 | 4 | | 57 |

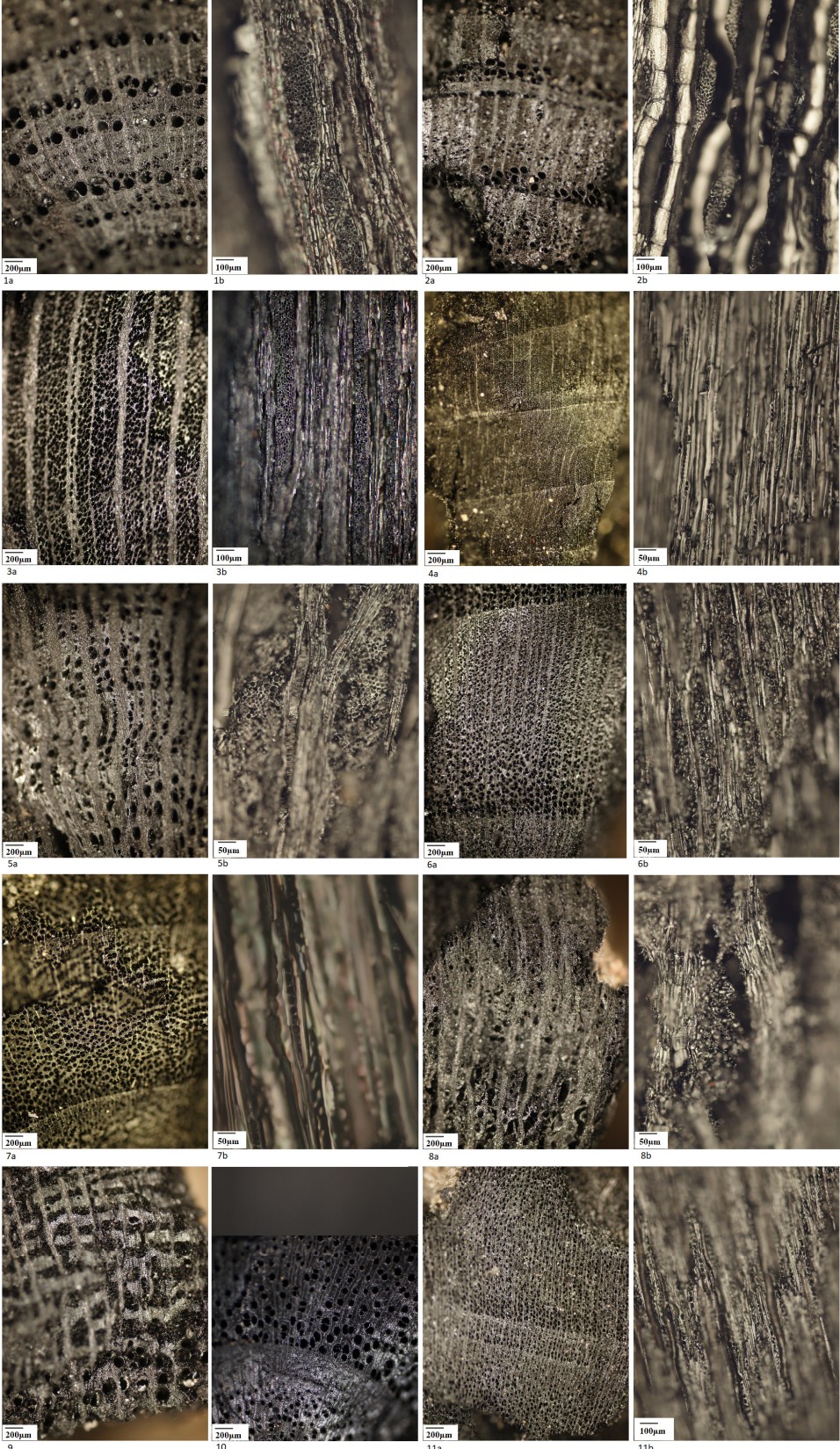

**Figure 7.** Vegetal taxa documented in Termez. 1: *Celtis australis*; 2: *Elaeagnus* sp.; 3: *Platanus orientalis*; 4: *Juniperus* sp.; 5: *Vitis vinifera*; 6: *Prunus* sp.; 7: Salicaceae; 8: *Tamarix* sp.; 9: *Ulmus* sp.; 10: *Fraxinus* sp.; 11: Rosaceae/Maloideae; a: cross section; b: tangential section.

### 3.3. Fauna

A number of 3762 faunal remains have been studied from Termez. Of these 1318 (35% of the NR) were determined to species or family levels. In general, the livestock at Termez was dominated by ovicaprids (>65% of the NISP), and these were the basis of the farming economy from the Greco-Bactrian period to the Islamic age (Table 2)

**Table 2.** Number of remains (NR), Number of Identified Specimens (NISP), and Minimum Number of Individuals (MNI) of the faunal remains in the different phases of occupation of Termez.

| Specie | Phase 1 | | | Phase 2 | | | Phase 3 | | |
|---|---|---|---|---|---|---|---|---|---|
| | NR | NISP | MNI | NR | NISP | MNI | NR | NISP | MNI |
| *Bos taurus* | 18 | 18 | 3 | 88 | 88 | 6 | 5 | 5 | 1 |
| *Ovis aries* | 31 | 31 | 6 | 38 | 38 | 8 | 52 | 52 | 5 |
| *Capra hircus* | 11 | 11 | 3 | 70 | 70 | 4 | 13 | 13 | 2 |
| Caprinae | 159 | 159 | 7 | 291 | 291 | 21 | 287 | 287 | 12 |
| *Gazella subgutturosa* | 2 | 2 | 1 | 5 | 5 | 2 | 1 | 1 | 1 |
| *Sus scrofa* | 5 | 5 | 2 | 46 | 46 | 5 | 3 | 3 | 2 |
| *Equus caballus* | 7 | 7 | 2 | 17 | 17 | 3 | 1 | 1 | 1 |
| *Equus asinus* | 1 | 1 | 1 | 6 | 6 | 2 | 1 | 1 | 1 |
| *Equus* sp. | 1 | 1 | 1 | 14 | 14 | 2 | - | - | - |
| Camelidae | - | - | - | 1 | 1 | 1 | - | - | - |
| *Saiga tatarica* | 1 | 1 | 1 | - | - | - | - | - | - |
| *Cervus elaphus* | 1 | 1 | 1 | 11 | 11 | 2 | - | - | - |
| *Canis l. familiaris* | 1 | 1 | 1 | 3 | 3 | 2 | 1 | 1 | 1 |
| *Vulpes corsac* | - | - | - | 1 | 1 | 1 | - | - | - |
| Leporidae | - | - | - | 1 | 1 | 1 | 15 | 15 | 1 |
| *Gallus g. domesticus* | 6 | 6 | 2 | 16 | 16 | 3 | 6 | 6 | 1 |
| *Columba livia/rupestris* sp. | - | - | - | - | - | - | 30 | 30 | 2 |
| *Corvus core/frugileus* sp. | - | - | - | - | - | - | 1 | 1 | 1 |
| *Anas/Mareca* sp. | - | - | - | - | - | - | 1 | 1 | 1 |
| *Passer* sp. | - | - | - | - | - | - | 18 | 18 | 2 |
| Indeterminable bird | - | - | - | - | - | - | 5 | 5 | 1 |
| Cyprinidae | - | - | - | - | - | - | 26 | 26 | 1 |
| Large size mammal | 52 | - | 2 | 138 | - | 4 | 43 | - | 3 |
| Medium size mammal | 16 | - | 2 | 102 | - | 4 | 4 | - | 2 |
| Small size mammal | 194 | - | 4 | 467 | - | 8 | 312 | - | 4 |
| Indeterminable | 377 | - | - | 534 | - | - | 205 | - | - |
| Total | 883 | 244 | 39 | 1849 | 608 | 79 | 1030 | 466 | 45 |

In the Greco-Bactrian and Yuezhi periods, sheep and goats were by far the most common domestic animals (82.37%), followed by cattle (7.38%) and equids (2.87%), which included both horses and donkeys. Poultry (2.46%) and, to a lesser extent, pigs (2.05%) were other sources of food. Dogs are represented by a single remain. Apart from the equids, all the domestic species display butchery marks indicating that their meat was consumed at the site. Wild species, like red deer, saiga antelope, and gazelle, appear in a very low proportion (<1%). These animals would live in the steppe areas near Termez,

where they were hunted and consumed, as butchery marks have been observed on their bones, particularly in the case of the gazelle.

During the Kushan and Kushano-Sasanian periods of occupation, sheep and goats still dominated the livestock herds but in a smaller proportion than before (65.62%). Cattle and pigs increase their representation (14.47% and 7.57%, respectively), even if they require a larger amount of food and water than sheep and goats; therefore, their increasing presence in Termez may be associated with greater availability of vegetal and water resources in this period. Horses and donkeys continue appearing in low percentages, like poultry and dogs (<3%). In this period appears the only camelid remains documented at the site. Except for the donkey, all the other domestic species display cut marks related to the meat consumption of those animals. Hunting was still a source of food in the Kushan and Kushano-Sasanian periods, as attested by the remains of red deer, gazelles, and leporids, some of them with cut marks on their bones. A remain of corsac fox was also documented among the wild fauna.

In the Islamic period, sheep and goats again predominated (75.54%). The percentages of these animals increase compared to the previous period, while cattle and swine decrease (<2%). Equids, poultry, and dogs still appear in small percentages. As regards hunting, while no red deer or saiga antelopes have been documented in this period, gazelles were hunted, and their bones display cut marks. A wide range of birds include ducks, pigeons, rooks/crows, and sparrows, but no cut marks are seen on their skeletal remains. Fishing is also attested in this period by the presence of cyprinids caught in the Amu Darya.

In the sheep and goats' group, most specimens are young adults less than two years old in all the periods at the site. The consumption of animals of this age would be because the meat of young individuals is more tender. The importance of adult individuals in the record seems to indicate they were kept for their milk and wool. In contrast, cattle were consumed at an adult age, which may be because they were used for their milk and secondary products and as draught animals in agriculture and transport. The consumption of beef is attested in the whole occupation of Termez except in the Islamic period. Pigs, also present in all the periods, display an age of death with a predominance of infantile and juvenile individuals, which indicates consumption of their meat at those young ages. Swine were most common in the Kushan and Kushano-Sasanian periods. Equids, equally present throughout the occupation, are represented by horses and donkeys, the latter in low percentages. A series of remains could not be assigned to one or another animal and might correspond to hybrid animals. The mortality profiles indicate a more significant proportion of adult animals, so they may have been used as draught animals. However, some remains of immature individuals display butchery marks, so their meat may have been consumed. Dogs appear in the whole occupation sequence, which is common in a situation of animal husbandry. Anthropic processing of those animals has been documented in the Greco-Bactrian and Yuezhi periods, and therefore their meat or skins may have been used. This would not be unusual, as it has been documented at other sites dated in Antiquity [96,97].

Birds appear in all periods but never in large numbers. Cockerels and hens are represented the most and are the only ones with marks of anthropic processing. Their mortality profiles indicate the consumption of poultry at an old age, which can be related to their use to obtain eggs.

The economic exploitation of domestic livestock was complemented by hunting and fishing in the surroundings of Termez. Gazelles, red deer, and saiga antelopes are well-adapted to steppe conditions. Although anthropic marks have only been detected on the remains of gazelles, red deer, and saiga, antelopes were also consumed historically at different sites in central Asia [95,98–102]. Fishing is documented in the Islamic period by the remains of cyprinids, indicating the use of fluvial resources in the Amu Darya.

## 4. Discussion

The natural conditions in the lower Surkhan Darya valley made this area attractive for human occupation. The region's aridity was not an obstacle to the development and growth of Termez since its foundation in the late 4th century BCE. The urban surface area expanded considerably from the small enclosure of the Citadel, the centre of the Greek city, over the 1500 years of its existence. Archaeological data reveal that the greatest urban expansion in Termez occurred in two periods; the first between the change of era and the early 2nd century, under the rule of the Kushan kings; the second in the early Islamic period (in the 8th century). During the Kushan period, the Citadel, Tchingiz Tepe and Shahristan were urban centres, and the northern suburb (Military Camp) was also occupied. The activity was probably intense in other peripheral parts of Termez between the change of era and the late 6th century, especially in the area where the Rabad would grow later. In the Islamic period, the city's urban growth took place early. Radiocarbon dates obtained for the Potters' Quarter demonstrate that some workshops were active as early as the 8th–9th centuries. The Citadel, Shahristan, Rabad, and both northern suburbs were occupied, while Tchingiz Tepe was abandoned before the Arab conquest of the region.

Urban development at Termez gradually adapted to the constant topographic changes caused by the accumulation of deposits derived from both the collapse of buildings and the accumulation of geological materials from the frequent sandstorms in southern Uzbekistan [103]. In some parts of the city, the ground level rose because of the accumulation of (aeolian) sand brought by the wind and the periodic collapse and deposition of the building structures made of adobe and rammed earth. This is especially attested in the thick stratigraphy found in the Citadel, with levels over 14 m thick in some places [14]. The Citadel seems to have been the urban centre throughout the historical sequence, as it maintained a population from its foundation to its destruction in 1220. The urban evolution within the fortified enclosure of Tchingiz Tepe also entailed managing the sedimentation of aeolian sand deposits and the collapse of adobe buildings; this is evidenced in some areas within the fortress where depositions comprising stratigraphies 2.5 m thick were formed by those processes [7,13]. However, in this case, the enclosure was abandoned by its inhabitants during the 5th century CE, like the Kara Tepe and Fayaz Tepe Buddhist monasteries located in the westernmost sector of ancient Termez. The expansion of the city towards the east from the 8th century compensated for that loss of inhabited area by occupying the only free space: the interior alluvial plain.

Human occupation was compatible with the conservation of the natural environment, especially the riparian woodland. This area supplied fuel to the population of Termez for domestic purposes (heat and cooking), craft work (especially pottery but also glass and metals), or to obtain timber for buildings or other uses. The anthropisation of the environment is attested by the introduction of allochthonous woody species from at least the 2nd–5th centuries onwards. Part of the farmland near the city must have been irrigated, as some of the plant species would have been cultivated, such as the fruit trees and grapevines: species that can only be grown in an arid environment with a supplementary water supply. Domestic animals varied over time, although ovicaprids are predominant in all periods. Pigs, which are adapted to an orchard environment, appear throughout the sequence, but are scarce in the Middle Ages, when they would not be kept because of the prohibition on their consumption by Islam.

In Termez, judging from the anthracological and faunal record, the diversified agricultural exploitation, including species claiming irrigated crops, coexists with a livestock exploitation including all domestic species. From the data obtained in the study, there are hardly any significant variations in vegetation or livestock over the period studied. Or, in other words, the variations observed are less relevant than the continuity in the patterns of resource exploitation. In Termez, it has also not been possible to discern adaptations to the climatic variations detected in the historical sequences, unlike in Juuku Valley and Targaz, two areas around Lake Issyk-Kul (Kyrgyzstan) that have recently been the subject of detailed analyses [104–106]. In these regions, a system of agropastoralism (combined

mountain agriculture) has been detected. This system is characterized by taking advantage of the variations in ecotones offered by the vertical gradient, an aspect that is especially important in the Juuku Valley.

However, the Termez records may not be incompatible with those obtained in the Issyk-Kul lake basin. The Issyk-Kul study areas have large variations due to the vertical gradient, resulting in diverse ecotones that allow for complementary economic use. In Juuku Valley, the vertical gradient makes the ecotones particularly sensitive to climatic variations that occurred during the Holocene. This probably means that the more visible variations in farming and herding strategies adapted to changes in vegetation at altitude. Termez is located in an oasis zone that benefits from the inputs of the Surkhan Darya to develop an irrigated crop agriculture. This irrigated agriculture seems to be an established fact, at least since Greek times. The faunal record of Termez presents some variations in the sequence, among which two stand out: the presence of wild species during the Greek-Yue zhi period (although in very low percentages) and the decline of the pig herd during the Islamic period (related to the religious prescription prohibiting the consumption of pork). However, judging from the data as a whole, these variations in the utilization of animal resources are less remarkable than the continuationist aspects.

The fauna and vegetation records detected in Termez are those of a large urban settlement in which agriculture and livestock coexist throughout the historical sequence. It would be necessary to carry out studies in the upper areas of the Surkhan Darya valley to detect which were the strategies that the human groups carried out in the peripheral spaces with respect to the settlement, in terms of agricultural and livestock use.

Termez was the most important city in northern Bactria (south of Uzbekistan) and prospered for longer than other cities founded at a similar time. About 20km to the west of Termez is Kampyr Tepe. According to the characteristics of the pottery found in the earliest levels, the foundation has been dated to about 300 BCE [107], or probably a little later, and posterior to the foundation of the Citadel in Termez [42,108]; however, some 14C chronologies suggest that a pottery workshop located within the Citadel of Kampyr Tepe was active in the late 4th century BCE [109]. The urban centre expanded to the north and east (low city) during the Greco-Bactrian, Yuezhi, and Kushan periods; it was abandoned in the late 2nd century CE [41], possibly because of a change in the course of the Amu Darya. Another relevant settlement in the Greek and Kushan periods was Dalverzin Tepe, 105km north of Termez, in the Shurkhan Darya valley. Its decline began at the end of the Kushan period (late 2nd century CE), although the population remained in specific areas of the urban layout until the 4th century. It was abandoned definitively in the 7th century; therefore, there is no evidence of Islamic occupation [110]. Khalchayan, located about 25km north of Dalverzin Tepe, was a centre founded in the Yuezhi period (mid-second century BCE) that became an important Kushan city; however, it was no longer inhabited in the Middle Ages as no Islamic levels have been documented [87,111]. Finally, Zar Tepe was a small city situated 15km north-west of Termez. It has provided a large pottery assemblage of the Kushan and Kushano-Sasanian periods. Its foundation has been dated to about the change of era, and it maintained a large population until the 4th century. Similar to other urban centres in the Surkhan Darya valley, the occupation seems to have been residual from that period, and it was depopulated in the Middle Ages [45].

The city of Termez, which was able to exist in a relatively hostile natural environment, was partially abandoned because of a catastrophic human action: the siege and destruction carried out by Genghis Khan in 1220. In fact, the Mongol conquest of central Asia had a powerful impact on the environment in the region, especially in the Amu Darya basin and the Aral Sea. The drop in the water level and the consequent regression detected in the 13th and 14th centuries might have resulted from the destruction of dams that controlled the tributaries of the Amu Darya by the Mongol armies. The impact would have been so significant as to change the course of the Amu Darya towards the Caspian Sea, leaving the Aral Sea without a supply of water [112]. Ancient Termez, which had outlived conquest by the Yuezhi nomads, the Sassanids and the Islamic forces, did not survive the destruction

caused by the Mongol army. However, it was re-founded soon after, just 10km to the south of the old centre of the Citadel.

## 5. Conclusions

The populations that inhabited the lower valley of the Surkhan Darya benefited from the water supply by this tributary of the Amu Darya to prosper in a hostile environment characterised by extreme summer and winter temperatures and low rainfall. The remains of fauna and charred wood reveal the use of different species, including possible fruit trees and domestic livestock; they comprise practically all the biotopes around the city in both Antiquity and the Middle Ages. The botanical data allows a reconstruction of the landscape with quite extensive river-bank woodland until c. 1300, probably denser than at present, especially in the area around the city, which is now quite barren. The hunted wild animals are characteristic of a steppe environment, which in the antique and medieval periods was little altered by human action. Gazelles, saiga antelopes, and red deer were hunting prey for the inhabitants of Termez.

The city experienced continuous growth from its foundation c. 300 BC until its destruction in 1220. The urban expansion of Termez was mainly inland. The city was able to adapt to the topographic changes caused by the accumulation of deposits created by the collapse of structures and the aeolian contributions of sand. The two most expansive moments of urban growth are detected in the Kushan period (early 1st century AD and mid-third century) and in the early Islamic period (8th and 9th centuries). Only the urban enclosure of Tchinguiz Tepe was abandoned before the Islamic conquest, probably because of maintenance problems due to the accumulation of sand deposits.

**Author Contributions:** Conceptualization, E.A., P.U., J.A., V.M.-F. and J.M.G.; Formal analysis, E.A., P.U., J.A., R.P. (Raquel Piqué), R.P. (Rodrigo Portero) and J.M.G.; Investigation, E.A., P.U., J.A., R.P. (Raquel Piqué), R.P. (Rodrigo Portero), V.M.-F. and J.M.G.; Methodology, E.A., P.U., J.A., R.P. (Raquel Piqué), R.P. (Rodrigo Portero), V.M.-F. and J.M.G.; Software, P.U. and J.A.; Visualization, E.A., P.U., J.A., R.P. (Raquel Piqué), R.P. (Rodrigo Portero) and J.M.G.; Writing—original draft, E.A., P.U., J.A., R.P. (Raquel Piqué), R.P. (Rodrigo Portero), V.M.-F. and J.M.G.; Writing—review & editing, E.A., P.U., J.A., R.P. (Raquel Piqué), R.P. (Rodrigo Portero), V.M.-F. and J.M.G. All authors have read and agreed to the published version of the manuscript.

**Funding:** This research has been financed by three projects: 'Caracterización arqueológica y arqueométrica de cerámicas de Asia central. Producción, distribución y tecnología' ('Archaeological and archaeometric characterisation of ceramics from Central Asia. Production, distribution and technology'), funded by MCIN/AE (PID2020-114096GB-C21) (IP V. Martínez Ferreras), 'Arqueología de los centros de producción cerámica y de consumo en Asia central. Una aproximación desde la teledetección' ('Archaeology of ceramic production and consumption centrers in Central Asia. A remote sensing approach'), funded by MCIN/AE (PID2020-114096GA-C22) (IP Paula Uribe Agudo) and 'Termez en Bactriana (Uzbekistán): colonia griega, centro del budismo centroasiático y ciudad islámica' ('Termez in Bactriana (Uzbekistan): Greek colony, Buddhish centre and Islamic city'), funded by the Palarq Foundation (Paleontology and Archaeology) (IP J.M. Gurt Esparraguera).

**Data Availability Statement:** Not applicable.

**Acknowledgments:** Special thanks to S. Pidaev from the Academy of Sciences of the Republic of Uzbekistan for his great help during the campaign in Uzbekistan.

**Conflicts of Interest:** The authors declare no conflict.

## Appendix A

**Table A1.** Analysed samples and [14]C results. Column A: Urban space, Excavation area, SU (Stratigraphic Unit); Colunm B: Dating material reference; Column C: Radiocarbon date code assigned by this Laboratory; Column D: Radiocarbon date with uncertainty expressed in terms of standard deviation; Column E: Calibrated date intervals centred on the modes of probability distribution for the true calibrated date corresponding to a total probability of 95.4% (range σσ) and the probability associated with each interval, respectively. In this set of intervals there is a probability of 95.4% that the true calibrated date will be found; Column F: Calibrated date intervals centred on the modes of probability distribution for the true calibrated date corresponding to a total probability of 68.3% (range σ) and the probability associated with each interval, respectively. In this set of intervals, there is a probability of 68.3% that the true calibrated date will be found.

| A | B | C | D | E | F |
|---|---|---|---|---|---|
| Citadel, Tower 2 | Charcoal | UBAR-1228 | 955 ± 30 BP | (95.4%) 1027–1160 cal AD | (11.2%) 1035–1049 cal AD<br>(57.1%) 1081–1152 cal AD |
| *Shahristan*, Wall | Bone | Beta-504292 | 1960 ± 30 BP | (7.9%) 42 cal BC–8 cal BC<br>(87.5%) 2 cal BC–130 Cal AD | (51.7%) 23–84 cal AD<br>(16.6%) 95–116 cal AD |
| *Shahristan*, Workshop 5, Kiln 3 | Charcoal | Beta-511510 | 560 ± 30 BP | (48.6%) 1312–1362 cal AD<br>(46.9%) 1387–1428 cal AD | (34.9%) 1326–1351 cal AD<br>(33.4%) 1394–1415 cal AD |
| Tchingiz Tepe, Sector RB1, SU 5 | Charcoal | UBAR-989/<br>CNA-366 | 2130 ± 45 BP | (18%) 356–280 cal BC<br>(0.2%) 253–250 cal BC<br>(76.7%) 232–41 cal BC<br>(0.5%) 9–2 cal BC | (6.6%) 340−325 cal BC<br>(54.1%) 199−95 cal BC<br>(7.6%) 73−56 cal BC |
| Tchingiz Tepe, Sector RB2, SU10 | Bone | UBAR-1144/<br>CNA-1190 | 2075 ± 40 BP | (95.4%) 197 cal BC–23 cal AD | (66.4%)153–42 cal BC<br>(1.9%) 8–4 cal BC |
| Tchingiz Tepe, Sector RB2, SU11 | Charcoal | UBAR-1145/<br>CNA-1189 | 2095 ± 55 BP | (9%) 353–286 cal BC<br>(0.7%) 228–218 cal BC<br>(85.1%) 211 cal BC–28 AD<br>(0.8%) 46–58 cal AD | (2.3%)195–187 cal BC<br>(64.1%) 178–42 cal BC<br>(1.8%) 8–2 cal BC |
| Tchingiz Tepe, Sector RB3, SU2 | Bone | UBAR-1241/<br>CNA-2253 | 1905 ± 35 BP | (3.1%) 28–45 cal AD<br>(92.4%) 58–225 cal AD | (8.5%) 84–96 cal AD<br>(59.8%) 115–204 cal AD |
| Tchingiz Tepe, Sector RB3, SU6 | Bone | UBAR-1240/<br>CNA-2252 | 1910 ± 35 BP | (95.4%) 26–219 cal AD | (12.2%) 80–99 cal AD<br>(56%) 109–204 cal AD |
| Tchingiz Tepe, Sector RC, SU10 | Charcoal | UBAR- 990 | 1940 ± 130 BP | (2.1%) 350–307 cal BC<br>(93.4%) 208–AD 405 cal BC | (3%) 95−74 cal BC<br>(65.2%) 56 cal BC–243 cal AD |
| Tchingiz Tepe, Sector RA, SU17 | Charcoal | UBAR-1146 | 1670 ± 55 BP | (13.7%) 250–295 cal AD<br>(81.7%) 311–540 cal AD | (9.5%) 258–281 cal AD<br>(53.2%) 331–435 cal AD<br>(3.3%) 518–529 cal AD |
| Tchingiz Tepe, Sector RC, SU2 | Charcoal | UBAR-1043 | 1680 ± 35 BP | (15.1%) 252–290 cal AD<br>(77.2%) 321–436 cal AD<br>(1%) 465–475 cal AD<br>(0.7%)500–509 cal AD<br>(1.6%) 515–531 cal AD | (5.5%) 265–272 cal AD<br>(62.8%) 351–417 cal AD |
| Tchingiz Tepe, Sector RC, SU3 | Charcoal | UBAR-1044 | 1760 ± 35 BP | (95.4%) 232–401 cal AD | (14.2%) 244–262 cal AD<br>(54%) 276–344 cal AD |
| Tchingiz Tepe, Sector RC, SU4 | Charcoal | UBAR-1045 | 1740 ± 50 BP | (95.4%) 224–417 cal AD | (28.1%) 250–295 cal AD<br>(40.2%) 310–376 cal AD |
| Tchingiz Tepe, Sector RC, SU5 | Charcoal | UBAR-1046 | 1755 ± 45 BP | (95.4%) 221–408 cal AD | (15%) 242–366 cal AD<br>(53.3%) 372–353 cal AD |
| Tchingiz Tepe, Sector RF, SU20 | Charcoal | UBAR-1053 | 1690 ± 230 BP | (0.6%) 346–316 cal BC<br>(94.3%) 204 cal BC–775 cal AD<br>(0.4%) 791–821 cal AD | (68,3%) 120–602 cal AD |
| Military Camp, Sector AC1, SU12 | Bone | UBAR-1242/<br>CNA-2254 | 1955 ± 35 BP | (95.4%) 51 cal BC–217 cal AD | (7.1%) 35–15 cal BC<br>(61.2%) 5–127 cal AD |

**Table A1.** *Cont.*

| A | B | C | D | E | F |
|---|---|---|---|---|---|
| Military Camp, Sector AC2, SU20 | Bone and Charcoal | UBAR-1125/ CNA–1121 UBAR-1126/ CNA 1122 | 1940 ± 25 BP | (92.9%) 12–169 cal AD (2.5%) 185–203 cal AD | (68.3%) 24–83 cal AD |
| Military Camp, Sector AC2, SU33 | Bone | UBAR-1127/CNA-1123 | 2155 ± 40 BP | (30.6%) 359–276 cal BC (2.2%) 261–244 cal BC (62.7%) 235–52 cal BC | (21.7%) 350–307 cal BC (46.6%) 208–106 cal BC |
| Military Camp, Sector AC2, SU12 | Bone | Beta-502182 | 1170 ± 30 BP | (73.9%) 772–901 cal AD (21.6%) 916–974 cal AD | (9.7%) 776–788 cal AD (49%) 825–894 cal AD (9.6%) 928–945 cal AD |
| Military Camp, Sector AC2, SU18 | Bone | Beta-515844 | 1000 ± 30 BP | (57%) 992–1051 cal AD (38.5%) 1080–1154 cal AD | (51.3%) 994–1045 cal AD (5.2%) 1085–1093 cal AD (11.7%) 1105–1121 cal AD |
| Potters quarter, Workshop 2, Kiln 2 | Charcoal | Beta-511511 | 1140 ± 30 BP | (4.9%) 774–787 cal AD (8.5%) 828–860 cal AD (82%) 870–992 cal AD | (16.1%) 883–903 cal AD (52.2%) 915–976 cal AD |
| Potters quarter, Workshop 11, Tannur | Charcoal | Beta-511509 | 1180 ± 30 BP | (82.7%) 771–900 cal AD (12.8%) 917–973 cal AD | (12.1%) 775–791 cal AD (56.2%) 821–891 cal AD (11.7%) 777–792 cal AD |
| Potters quarter, Workshop 11, Kiln 1, SU1 | Charcoal | Beta-545671 | 1150 ± 30 BP | (7.5%) 773–789 cal AD (88%) 824–988 cal AD | (4.8%) 777–785 cal AD (5.4%) 835–846 cal AD (17.7%) 877–901 cal AD (40.3%) 916–974 cal AD |
| Potters quarter, Workshop 11, Kiln 1, SU1 | Charcoal | Beta-502183 | 1190 ± 30 BP | (1.6%) 709–722 cal AD (88%) 771–897 cal AD (5.8%) 923–952 cal AD | (11.4%) 777–791 cal AD (1.6%) 805–807 cal AD (55.3%) 820–886 cal AD |
| Potters quarter, Workshop 11, Kiln 3, SU21 | Charcoal | Beta-545670 | 1190 ± 30 BP | (1.6%) 709–722 cal AD (88%) 771–897 cal AD (5.8%) 923–952 cal AD | (11.4%) 777–791 cal AD (1.6%) 805–807 cal AD (55.3%) 820–886 cal AD |
| Potters quarter, Workshop 11, SU32 | Charcoal | Beta-550228 | 1120 ± 30 BP | (1.8%) 774–785 cal AD (1.4%) 833–846 cal AD (92.3%) 876–995 cal AD | (34.1%) 893–933 cal AD (34.2%) 940–977 cal AD |

**Table A2.** Quantification of ceramic productions recovered in surface surveying.

| Ware \ Square | Sha1 | Sha1% | Sha2 | Sha2% | Rabad1 | Rabad1% | KBTK1 | KBTK1% |
|---|---|---|---|---|---|---|---|---|
| Red-figure ware (Greek) | 1 | 0.02 | | | | | | |
| White-slipped ware (Greek) | 1 | 0.02 | | | | | | |
| Black-slipped tableware (Greek) | 3 | 0.06 | | | | | 2 | 0.12 |
| Total Greek pottery | 5 | 0.11 | | | | | 2 | 0.12 |
| Grey pottery (Yuezhi) | 6 | 0.13 | 2 | 0.05 | | | | |
| Total grey pottery | 6 | 0.13 | 2 | 0.05 | | | | |
| Slipped tableware (Kushan/Sassanian) | 154 | 3.29 | 359 | 8.94 | 60 | 3.93 | 22 | 1.31 |
| Common ware (Kushan/Sassanian) | 13 | 0.28 | 12 | 0.30 | 5 | 0.33 | 16 | 0.95 |
| Common cooking ware (Kushan/Sassanian) | | | 11 | 0.27 | 2 | 0.13 | 1 | 0.06 |
| Total Kushan/Sassanian pottery | 167 | 3.57 | 382 | 9.52 | 67 | 4.39 | 39 | 2.32 |
| Common cooking ware, black slip partially covering the surface (Islamic) | 186 | 3.98 | 76 | 1.89 | 39 | 2,55 | 28 | 1.66 |

**Table A2.** *Cont.*

| Ware \ Square | Sha1 | Sha1% | Sha2 | Sha2% | Rabad1 | Rabad1% | KBTK1 | KBTK1% |
|---|---|---|---|---|---|---|---|---|
| Unglazed ware, comb-impressed/comb-incised decoration (Islamic) | 25 | 0.53 | 15 | 0.37 | 6 | 0.39 | 38 | 2.26 |
| Unglazed ware, applied decoration (Islamic) | 1 | 0.02 | | | | | 1 | 0.06 |
| Unglazed ware, stamped relief decoration (Islamic) | 10 | 0.21 | 6 | 0.15 | 3 | 0.20 | | |
| Unglazed ware, moulded relief decoration (Islamic) | 11 | 0.24 | 5 | 0.12 | 2 | 0.13 | 5 | 0.30 |
| Glazed white ware (Islamic) | 96 | 2.05 | 88 | 2.19 | 62 | 4.06 | 13 | 0.77 |
| Glazed green monochrome ware (Islamic) | 18 | 0.39 | 6 | 0.15 | 11 | 0.72 | 6 | 0.36 |
| Glazed turquoise monochrome ware (Islamic) | 65 | 1.39 | 36 | 0.90 | 30 | 1.96 | 5 | 0.30 |
| Glazed yellow monochrome ware (Islamic) | 4 | 0.09 | 3 | 0.07 | 1 | 0.07 | | |
| Underglaze painted ware/slip painted (Islamic) | 21 | 0.45 | 16 | 0.40 | 7 | 0.46 | 18 | 1.07 |
| Splashed *sgraffiato* ware (Islamic) | 50 | 1.07 | 37 | 0.92 | 20 | 1.31 | 24 | 1.43 |
| Figured/geometric decoration glazed ware (Islamic) | 122 | 2.60 | 34 | 0.84 | 20 | 1.31 | 5 | 0.30 |
| Glazed overfired (not specific) (Islamic) | 11 | 0.24 | 1 | 0.02 | 5 | 0.33 | | |
| Glazed altered (not specific) (Islamic) | | | | | 1 | 0.07 | | |
| Sphero-conical vessel (Islamic) | 35 | 0.75 | 23 | 0.57 | 7 | 0.46 | 38 | 2.26 |
| Total Islamic ware | 655 | 14.01 | 346 | 8.62 | 214 | 14.01 | 181 | 10.75 |
| Common ware | 3593 | 76.87 | 3187 | 79.40 | 1211 | 79.31 | 1420 | 84.32 |
| Common cooking ware | 248 | 5.31 | 97 | 2.42 | 35 | 2.29 | 42 | 2.49 |
| Total undetermined pottery | 3841 | 82.18 | 3284 | 81.81 | 1246 | 81.60 | 1462 | 86.82 |
| Total | 4674 | 100 | 4014 | 100 | 1527 | 100 | 1684 | 100 |

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
