# Peer review of "Adaptive Dynamics of Settlement Models in the Urban Landscape of Termez (Uzbekistan) from c. 300 BCE to c. 1400 CE"

_land, doi:10.3390/land12081550_

Round 1
Reviewer 1 Report
This study presents the dynamic occupation of the Termez site since the prehistorical time. The authors used multi proxies to analyze and determine the time and groups who occupied the area. Overall, this study is well presented and the objectives are clear. However, there are a few questions that the authors might consider.
1. The significance of the study was not presented in the introduction.
2. The introduction of this study is a historical and geographical introduction of the study area, but not an introduction to this study. I suggest reorganizing the introduction.
3. I think the authors should also state the current vegetation type in the introduction of the study area.
4. The subtitle “Fauna and Animal” seems redundant, since Fauna could be used to state the animal in the region.
5. Resolution of the figures needs to be improved.
Author Response
This study presents the dynamic occupation of the Termez site since the prehistorical time. The authors used multi proxies to analyze and determine the time and groups who occupied the area. Overall, this study is well presented and the objectives are clear. However, there are a few questions that the authors might consider.
- The significance of the study was not presented in the introduction. Done. A short presentation of has been included in Lines 50-60.
- The introduction of this study is a historical and geographical introduction of the study area, but not an introduction to this study. I suggest reorganizing the introduction. A short presentation of has been included in Lines 50-60.
- I think the authors should also state the current vegetation type in the introduction of the study area. Done. Some information on the topic has been included in Lines 115-123.
- The subtitle “Fauna and Animal” seems redundant, since Fauna could be used to state the animal in the region. Simplified in “Fauna”.
- Resolution of the figures needs to be improved. Quality images are attached in zip file.
Reviewer 2 Report
This article analyzes the process of formation and development of the ancient city of Termez, in the south of current Uzbekistan, in a wide period of time, between 300 B.C. until 1220 a.C. from a multidisciplinary perspective, what must be highlighted. The current city that bears the name of Termez does not correspond to the ancient city, separated by ten kilometers nowadays. The 1220 deadline of the study is because the city was besieged and destroyed by Genghis Khan. In addition, the Mongolian troops destroyed the dams of the tributaries of the Amu Darya River, which caused the drop of its level and changed the river's course towards the Caspian Sea, leaving the Aral Sea without its water supply.
This is a very interesting study, which is carried out from an enriching interdisciplinary perspective. However, the authors should refer to a state of the art on the urbanization process of the entire region in order to better understand what this study contributes to the knowledge of the development and decline of the ancient and medieval urban world in the south of present-day Uzbekistan. On the other hand, the conclusions are very poor and do not include the main contributions of the study. It is also necessary to include the future perspectives of this study in the coming years.
Author Response
This article analyzes the process of formation and development of the ancient city of Termez, in the south of current Uzbekistan, in a wide period of time, between 300 B.C. until 1220 a.C. from a multidisciplinary perspective, what must be highlighted. The current city that bears the name of Termez does not correspond to the ancient city, separated by ten kilometers nowadays. The 1220 deadline of the study is because the city was besieged and destroyed by Genghis Khan. In addition, the Mongolian troops destroyed the dams of the tributaries of the Amu Darya River, which caused the drop of its level and changed the river's course towards the Caspian Sea, leaving the Aral Sea without its water supply.
This is a very interesting study, which is carried out from an enriching interdisciplinary perspective. However, the authors should refer to a state of the art on the urbanization process of the entire region in order to better understand what this study contributes to the knowledge of the development and decline of the ancient and medieval urban world in the south of present-day Uzbekistan.
A short text has been added in order to present the historical and geographical context of the region (see Lines 63-85).
On the other hand, the conclusions are very poor and do not include the main contributions of the study. It is also necessary to include the future perspectives of this study in the coming years.
Discussion and Conclusion have been improved and expanded (see Lines 784-814 and 862-869).
Reviewer 3 Report
This study considers different and multidiscipline tools to analyze an interesting problem of settlement in the historical and archeological aspects. The manuscript is well-written, and its construction is correct. Figures and tables are visible and understandable. These are also correctly formatted. The good idea is to present some parts of the analyses in the appendix. Background, objectives, methodology, results, and findings are clear and appropriate.
The topic is correct for the “Land” journal and its special issue “Resilience in Historical Landscapes”. I detected a small but distinct aspect of the review that should be considered. I think that the study planned for publishing in this journal should be better connected with topics published there yet. I suggest enhancement of the literature base for the selected papers published in “Land”. Please, use your own keywords “Zooarchaeology”, “Anthropogenic dynamics” etc. to find fresh and representative papers worth citing.
New references will enlarge the basis for the study and should be also commented on in the discussion part of the manuscript. Especially, it should be shown what are the differences between the former studies and this one and what is new and important given the obtained results.
The smaller, technical remark: please add or make more visible the country borders on the lower part of figure 1.
Minor editing of English language required.
Author Response
This study considers different and multidiscipline tools to analyze an interesting problem of settlement in the historical and archeological aspects. The manuscript is well-written, and its construction is correct. Figures and tables are visible and understandable. These are also correctly formatted. The good idea is to present some parts of the analyses in the appendix. Background, objectives, methodology, results, and findings are clear and appropriate.
The topic is correct for the “Land” journal and its special issue “Resilience in Historical Landscapes”. I detected a small but distinct aspect of the review that should be considered. I think that the study planned for publishing in this journal should be better connected with topics published there yet. I suggest enhancement of the literature base for the selected papers published in “Land”. Please, use your own keywords “Zooarchaeology”, “Anthropogenic dynamics” etc. to find fresh and representative papers worth citing.
New references will enlarge the basis for the study and should be also commented on in the discussion part of the manuscript. Especially, it should be shown what are the differences between the former studies and this one and what is new and important given the obtained results.
New references related to the subject of study have been included, which have allowed comparative analyzes that we judge of interest. The discussion could have been significantly improved thanks to this indication.
The smaller, technical remark: please add or make more visible the country borders on the lower part of figure 1.
Done.
Comments on the Quality of English Language
Minor editing of English language required.
The text has been revised and edited to improve the quality of the English language.
Round 2
Reviewer 3 Report
My main remark was concerned with the poor connectivity of this manuscript with topics presented in “Land”. This remark was considered on a very small level. Only 3 new positions were added. Using the keywords “Anthropogenic dynamics” I identified 41 papers published in “Land” yet. Research background (literature review) and discussion should concerns not only territorial connections (that means the papers dedicated to Kyrgyzstan), but a broader scientific sound dedicated to exemplary “Anthropogenic dynamics”. I still think that this aspect is represented too small.
The map in figure 1 is corrected partially because the border between Tajikistan and Afghanistan is still not visible.
Author Response
We have carefully reviewed the contents of the 41 papers published in ‘Land’ grouped under the keyword ‘Anthropogenic dynamics’. All of them are studies on current anthropogenic landscapes. We have not been able to find a direct connection with the archaeological research we have carried out at Termez. To avoid confusion of content in the journal, we have decided to remove ‘Anthropogenic dynamics’ from the list of keywords.
Figure 1 has been corrected.